# Adaptive Personalized Federated Learning
## via Multi-task Averaging of Kernel Mean Embeddings

**Jean-Baptiste Fermanian** [1]   **Batiste Le Bars** [2]   **Aurélien Bellet** [1]

## Abstract

Personalized Federated Learning (PFL) enables a collection of agents to collaboratively learn individual models without sharing raw data. We propose a new PFL approach in which each agent optimizes a weighted combination of all agents' empirical risks, with the weights learned from data rather than specified a priori. The novelty of our method lies in formulating the estimation of these collaborative weights as a kernel mean embedding estimation problem with multiple data sources, leveraging tools from multi-task averaging to capture statistical relationships between agents. This perspective yields a fully adaptive procedure that requires no prior knowledge of data heterogeneity and can automatically transition between global and local learning regimes. By recasting the objective as a high-dimensional mean estimation problem, we derive finite-sample guarantees on local excess risks for a broad class of distributions, explicitly quantifying the statistical gains of collaboration. To address communication constraints inherent to federated settings, we also propose a practical implementation based on random Fourier features, which allows one to trade communication cost for statistical efficiency. Numerical experiments validate our theoretical results.

## 1. Introduction

Despite the growing volume of data collected worldwide, many application domains, such as medicine and ecology, remain fundamentally data-limited. In such contexts, a major scientific challenge is to learn effective models from data originating from multiple sources that are often heterogeneous and biased, yet share enough similarities to be jointly exploited. In healthcare, they may arise from different hospitals observing distinct patient populations or operating under variations in medical devices and clinical protocols (Rieke et al., 2020; Xu et al., 2021; Nguyen et al., 2022). Similar challenges also appear in astrophysics, where data are collected across instruments, wavelengths, or angular resolutions that vary between observation centers (Elmahallawy & Luo, 2022; Razmi et al., 2022; Chen et al., 2022).

When all data are directly accessible, learning from heterogeneous sources is commonly framed as multi-task learning (Caruana, 1997; Zhang & Yang, 2021). We instead consider a more challenging and increasingly relevant setting in which data are distributed across multiple data owners (e.g. hospitals), hereafter referred to as *agents*, that seek to collaborate without sharing their raw data. Such constraints may arise because data are too sensitive to be shared or due to transmission costs, bandwidth or storage limitations, institutional constraints, or legal barriers. Over the past few years, this setting has been addressed within the framework of Federated Learning (FL, Kairouz et al., 2021). Early FL methods primarily focused on learning a single global model that performs well on average across all decentralized data. However, due to the heterogeneity that naturally arises across agents in decentralized environments, this one-size-fits-all objective has quickly been recognized as a limitation. To address this issue, the paradigm of *Personalized* FL (PFL) has emerged. Similarly to multi-task learning, PFL aims to learn a agent-specific models while still enabling collaboration between them. The central challenge is then to manage inter-agent heterogeneity and control the bias induced by leveraging data from other agents.

Numerous methods have been proposed to address this problem (see Section 3 for an overview). Most of them enable collaboration by assuming some structure among agents, and in some cases even require that this structure is known—for example, that all local models are close to a global model, that agents form a fixed number of clusters, or that each model can be expressed as a linear combination of a local and a shared global model. These assumptions, however, are often violated in practice, limiting the meth-

---

[1]PreMeDICal, Inria, Idesp, Inserm, University of Montpellier [2]University of Lille, Inria, CNRS, Centrale Lille, UMR 9189, CRIStAL, F-59000 Lille. Correspondence to: Jean-Baptiste Fermanian <jean-baptiste.fermanian@inria.fr>.

*Proceedings of the 43$^{rd}$ International Conference on Machine Learning*, Seoul, South Korea. PMLR 306, 2026. Copyright 2026 by the author(s).

ods' effectiveness. More generally, existing approaches are largely heuristic and provide no generalization guarantees demonstrating a statistical benefit of collaboration over learning in isolation. In this work, we address these gaps by proposing an approach that requires no prior knowledge of agents' heterogeneity, automatically adapts to their underlying structure, and comes with generalization guarantees that quantify the advantage of collaboration. More precisely:

• We formulate the PFL problem as learning a mixture of the agents' data distributions. Assuming the loss lies to a *Reproducing Kernel Hilbert Space* (RKHS), we link the excess risk to the *Maximum Mean Discrepancy* (MMD) between a target agent's distribution and the estimated mixture (Eq. 3 and Lemma 4.3). Learning the mixture weights by minimizing the MMD then amounts to aggregating the *Kernel Mean Embeddings* (KMEs) of the agents' distributions.

• Since KMEs are high-dimensional means, we leverage the Q-aggregation mean estimation method of Blanchard et al. (2024) to estimate the mixture weights. A novel theoretical analysis (Theorem 4.4 and Corollary 4.6) demonstrates the resulting statistical gains in terms of the excess risk evaluated on the target agent's distribution.

• As sharing KMEs directly would typically require sharing all the raw data, conflicting with federated learning requirement, we propose a practical method based on *random Fourier features*, for which we derive theoretical guarantees quantifying the trade-off between communication costs and statistical efficiency (Theorem 5.2).

• We validate our approach empirically on synthetic and real-world data, showing that it effectively adapts to the heterogeneity across agents.

## 2. Preliminaries

**Notations.** For an integer $B \in \mathbb{N}$, let $[\![B]\!] := \{1, \ldots, B\}$. We denote by $\mathcal{S}_B := \left\{ \boldsymbol{\omega} \in [0,1]^B : \sum_{i=1}^B \omega_i = 1 \right\}$ the $B$-simplex. For $a, b \in \mathbb{R}$, $a \vee b := \max(a, b)$.

### 2.1. Setting and Objective

We consider a setting with $B$ agents, each having access to its own local dataset. The dataset of an agent $k \in [\![B]\!]$, denoted by $Z_\bullet^{(k)} = \left\{ Z_i^{(k)} \right\}_{i=1}^{n_k}$, consists of $n_k$ i.i.d. samples drawn from a distribution $\mathbb{P}_k$ with support in $\mathcal{Z}$. We assume no prior similarity between $\mathbb{P}_k$ and the distributions of other agents. Our goal is to leverage this multiplicity of sources to improve the model learned for a target agent, say $k = 1$, beyond what could be achieved using only its local data. This kind of setting is referred as *all-for-one* (Even et al., 2022). Formally, we study the excess risk of a learned model $\widehat{\theta}$:

$$\mathbb{E}\left[\mathcal{R}_1(\widehat{\theta})\right] - \mathcal{R}_1(\theta_1^*) , \tag{1}$$

where $\mathcal{R}_k(\theta) = \mathbb{E}_{Z \sim \mathbb{P}_k}[\ell_\theta(Z)]$ denotes the risk associated with a loss function $\ell_\theta : \mathcal{Z} \to \mathbb{R}$ and $\theta_k^* \in \arg\min_{\theta \in \Theta} \mathcal{R}_k(\theta)$ is an optimal model for agent $k$.

In a classical learning setting, we cannot directly minimize the population risk $\mathcal{R}_1$, so a standard approach is to minimize its empirical counterpart using the local dataset $Z_\bullet^{(1)}$. The empirical risk is defined as $\widehat{\mathcal{R}}_1(\theta) = \frac{1}{n_1} \sum_{i=1}^{n_1} \ell_\theta(Z_i^{(1)})$, and the excess risk (1) of its minimizer is well studied, typically scaling at $\mathcal{O}(n_1^{-1/2})$ (see e.g. Bach, 2024). In this work, we address the PFL problem by minimizing a weighted empirical risk instead:

$$\widehat{\mathcal{R}}_{\widehat{\boldsymbol{\omega}}}(\theta) := \sum_{k=1}^B \widehat{\omega}_k \widehat{\mathcal{R}}_k(\theta) , \tag{2}$$

where the weights $\widehat{\boldsymbol{\omega}} \in \mathcal{S}_B$ are themselves to be learned. Our goal is to show that minimizing this weighted risk can improve the excess risk compared to minimizing $\widehat{\mathcal{R}}_1$. We emphasize that, given the weights, our focus is not on minimizing (2) in a FL setting, many algorithms exist for that, but on computing the weights in $\widehat{\boldsymbol{\omega}}$ and deriving statistical bounds on the excess risk (1). Recall that, if the optimization is exact, i.e., $\widehat{\theta} \in \arg\min_\theta \widehat{\mathcal{R}}_{\widehat{\boldsymbol{\omega}}}(\theta)$, the excess risk is controlled by (twice) the generalization error:

$$\mathbb{E}\left[\mathcal{R}_{\widehat{\boldsymbol{\omega}}}(\widehat{\theta})\right] - \mathcal{R}_1(\theta_1^*) \leq 2\mathbb{E}\left[\sup_{\theta \in \Theta} \left|\widehat{\mathcal{R}}_{\widehat{\boldsymbol{\omega}}}(\theta) - \mathcal{R}_1(\theta)\right|\right] , \tag{3}$$

which is the central quantity analyzed in this work.

*Remark* 2.1 (Optimization error). In the case of imperfect minimization of $\widehat{\mathcal{R}}_{\widehat{\boldsymbol{\omega}}}$, an additional optimization error term, typically dependent on the regularity of the loss $\theta \mapsto \ell_\theta$ and the number of iterations, would be added to the generalization error in (3). However, controlling this is beyond the scope of this work, which focuses on the statistical error.

### 2.2. RKHS and KME

We briefly recall some tools from kernel methods, which are central to our approach. A kernel $\kappa : \mathcal{Z} \times \mathcal{Z} \to \mathbb{R}$ is positive definite if for any finite sequences $a_i \in \mathbb{R}$ and $z_i \in \mathcal{Z}$, $\sum_{ij} a_i a_j \kappa(z_i, z_j) \geq 0$. For any positive definite kernel, there exists a unique Hilbert space $\mathcal{H} \subset \{f, f : \mathcal{Z} \to \mathbb{R}\}$ such that for any $h \in \mathcal{H}$ and $z \in \mathcal{Z}$, $\langle h, \kappa(z, \cdot) \rangle_{\mathcal{H}} = h(z)$ (Aronszajn, 1950). This space is called a RKHS and corresponds to the completion of the span of $\{\phi_\kappa(z) : z \in \mathcal{Z}\}$, where $\phi_\kappa(z)(\cdot) = \kappa(z, \cdot)$ is called the feature map.

For a distribution $\mathbb{P}$ on $\mathcal{Z}$, the KME $\mu_{\mathbb{P}} \in \mathcal{H}$ (Smola et al., 2007) is defined as

$$\mu_{\mathbb{P}}(\cdot) = \mathbb{E}_{Z \sim \mathbb{P}}[\phi_\kappa(Z)] = \mathbb{E}_{Z \sim \mathbb{P}}[\kappa(Z, \cdot)] . \tag{4}$$

Then, for any function $h \in \mathcal{H}$, $\mathbb{E}_{Z \sim \mathbb{P}}[h(Z)] = \langle h, \mu_{\mathbb{P}} \rangle_{\mathcal{H}}$. KMEs allow defining a distance between distributions, the

MMD (Gretton et al., 2012):

$$\mathrm{MMD}_{\kappa}(\mathbb{P}, \mathbb{Q}) = \|\mu_{\mathbb{P}} - \mu_{\mathbb{Q}}\|_{\mathcal{H}}$$
$$= \sup_{\substack{h \in \mathcal{H}: \|h\|_{\mathcal{H}} = 1}} \mathbb{E}_{\substack{Z \sim \mathbb{P} \\ W \sim \mathbb{Q}}}[h(Z) - h(W)]. \quad (5)$$

Computing the MMD only requires computing scalar products between KMEs, which can easily be estimated. Indeed, one can notice that $\langle \mu_{\mathbb{P}}, \mu_{\mathbb{Q}} \rangle_{\mathcal{H}} = \mathbb{E}_{Z \sim \mathbb{P}, W \sim \mathbb{Q}}[\kappa(Z, W)]$, which is estimated by averaging kernel evaluations across pairs of samples from each distribution. In particular, the KME of $\mathbb{P}$ can be approximated by $\widehat{\mu}_{\mathbb{P}}(\cdot) = \frac{1}{n} \sum_{i=1}^{n} \kappa(Z_i, \cdot)$, where $(Z_i)_{i=1}^{n}$ are i.i.d. samples from $\mathbb{P}$. The mean squared error of this approximation corresponds to the MMD between the true and empirical distributions:

$$\mathbb{E}\left[\mathrm{MMD}_{\kappa}^2(\mathbb{P}, \widehat{\mathbb{P}})\right] = \mathbb{E}\left[\|\widehat{\mu}_{\mathbb{P}} - \mu_{\mathbb{P}}\|_{\mathcal{H}}^2\right] = \frac{\mathrm{Tr}\,\Sigma(\mathbb{P})}{n}, \quad (6)$$

where $\Sigma(\mathbb{P}) : \mathcal{H} \to \mathcal{H}$ is the covariance operator of the pushforward of $\mathbb{P}$ into $\mathcal{H}$ by $\phi_{\kappa}$ (see Definition E.3). Note that the trace and operator norm of such operator are defined similarly to those in finite dimension (see Definition E.4).

## 2.3. Random Fourier Features

Computing inner products between KMEs can be costly for large datasets. A standard approach to reduce this cost is to approximate the RKHS $\mathcal{H}$ using Random Fourier Features (RFF, Rahimi & Recht, 2007; 2008a;b).

Let $\kappa : \mathcal{Z} \times \mathcal{Z} \to \mathbb{R}$ be a translation-invariant positive definite kernel (i.e., $\exists \Phi : \kappa(x, y) = \Phi(x - y)$). By Bochner's theorem (Bochner, 2005), $\mathcal{H}$ with feature map $\phi_{\kappa}$ can be approximated by a finite-dimensional RKHS $\mathcal{H}_{\Gamma} \subset \mathbb{R}^D$ with mapping $\phi_{\Gamma}$, using a distribution $p$ on $\mathcal{Z}$. Specifically, draw $w_s \sim p$, $b_s \sim \mathcal{U}([0, 2\pi])$ for $s \in [\![D]\!]$, and define:

$$\phi_{\Gamma}(z) = \sqrt{\frac{2}{D}} \big( \cos(\langle z, w_s \rangle + b_s) \big)_{s=1}^{D} \in \mathcal{H}_{\Gamma}. \quad (7)$$

This mapping preserves the kernel in expectation:

$$\kappa(z, z') = \langle \phi_{\kappa}(z), \phi_{\kappa}(z') \rangle_{\mathcal{H}} = \mathbb{E}_{\Gamma}[\langle \phi_{\Gamma}(z), \phi_{\Gamma}(z') \rangle_{\mathbb{R}^D}], \quad (8)$$

where the expectation is taken over the random parameters $\Gamma = ((w_s, b_s))_{s=1}^{D}$. Although elements of $\mathcal{H}_{\Gamma}$ are represented as vectors in $\mathbb{R}^D$, they define functions from $\mathcal{Z}$ to $\mathbb{R}$: if $h \in \mathcal{H}_{\Gamma}$, then $h(z) = \langle h, \phi_{\Gamma}(z) \rangle_{\mathbb{R}^D}$. Other properties of these objects are detailed in Appendix E.

## 3. Related Work

**Personalized federated learning.** Addressing agent heterogeneity in FL through personalization has attracted significant interest in recent years, leading to a variety of approaches (Kulkarni et al., 2020; Smith et al., 2017; Sattler et al., 2020; Mansour et al., 2020; Dinh et al., 2020; Tan et al., 2022; Cui et al., 2022; Wu et al., 2023).

Many approaches rely on strong assumptions about the form of heterogeneity. Meta-learning, model interpolation, and fine-tuning methods, for instance, assume that a global model or shared representation provides a good starting point for all agents (Chen et al., 2018; Arivazhagan et al., 2019; Fallah et al., 2020; Li et al., 2021; Deng et al., 2020; Hanzely et al., 2020). Cluster-based methods assume that agents belong to a fixed number of clusters and learn one model per cluster (Sattler et al., 2020; Ghosh et al., 2020; Marfoq et al., 2021). Kernel-based methods similarly assume shared structure, either a shared model with personalized combinations of base kernels (Ghari & Shen, 2022) or a shared kernel with personalized models (Achituve et al., 2021). Such assumptions limit adaptability: if the underlying structure does not match these priors, these methods can fail entirely.

To overcome these limitations, a more flexible line of work learns similarity or collaboration weights between agents, allowing adaptation to diverse heterogeneity patterns. For example, Zantedeschi et al. (2020) regularize local objectives with pairwise weights $w_{ij}|\theta_i - \theta_j|$ that are learned jointly with the models, while CoBo (Hashemi et al., 2024) learns such weights dynamically via bilevel optimization. In a related direction, Kharrat et al. (2025) construct a collaboration graph that guides each client in selecting suitable collaborators. Our approach builds on this paradigm, focusing on the computation of collaboration weights in a statistically grounded manner.

Despite extensive work, few methods provide generalization guarantees. Most focus on the convergence of optimization algorithms on training data, without quantifying the excess risk over the agents' population distributions. Some obtain bounds under stochastic optimization with fresh samples at each iteration (Even et al., 2022; Scaman et al., 2024; Philippenko et al., 2025), limiting applicability and preventing multiple passes over datasets of varying sizes. Moreover, these bounds typically assume oracle knowledge of collaboration weights or consider weights learned for very specific models, such as linear regression. A few works, like Ding & Wang (2022), provide finite-sample generalization guarantees, but only for theoretical weights rather than weights learned in practice. In contrast, our approach establishes excess risk guarantees for weights actually estimated from data, without structural assumptions, explicitly demonstrating the statistical benefit of collaboration.

**High-dimensional multiple mean estimation.** Our contribution builds on the classical problem of multiple mean estimation. Stein's paradox famously showed that the standard empirical averages are suboptimal for estimating multiple means, and that shrinkage estimators can improve estima-

tion accuracy, especially in high-dimensional settings (Stein, 1956; James & Stein, 1961; Efron & Morris, 1972; George, 1986). More recently, this problem has been revisited in multi-task learning (Martínez-Rego & Pontil, 2013; Feldman et al., 2014; Duan & Wang, 2023) and in the estimation of KMEs (Muandet et al., 2014). From a high-dimensional perspective, Marienwald et al. (2021), extended in Blanchard et al. (2024), propose a KME aggregation method, which we leverage in this work. As detailed below, we relate the estimation of the weighted risk (2) to learning a mixture over agents. We then use KME aggregation to estimate the corresponding weights, and quantify the resulting improvement.

# 4. Personalized Learning as High-Dimensional Mean Estimation

In this section, we reformulate the PFL objective introduced in Section 2.1, namely, learning the weights $\widehat{\omega}$ to minimize the excess risk, as a high-dimensional mean estimation problem in a RKHS with multiple data sources. To the best of our knowledge, this is the first work establishing a formal connection between these two problems, which enables us to transfer algorithms with strong statistical guarantees from the latter setting to PFL. We emphasize that our focus here is on a general approach for learning the weights in $\widehat{\omega}$; the practical implementation in a federated learning context is deferred to Section 5. Proofs are provided in Section F.

## 4.1. Controlling Generalization with MMD

Recall from Eq. (2) that our goal is to leverage the data distributions of all agents to learn a better model for a target agent, say agent 1, by estimating weights $\widehat{\omega} \in \mathcal{S}_B$. Equivalently, the local empirical distribution $\widehat{\mathbb{P}}_1$ is replaced by a mixture of empirical distributions:

$$\widehat{\mathbb{P}}(\widehat{\omega}) = \sum_{k=1}^{B} \widehat{\omega}_k \widehat{\mathbb{P}}_k. \tag{9}$$

The weights $\widehat{\omega} \in \mathcal{S}_B$ are chosen so that this mixture better approximates the target distribution $\mathbb{P}_1$ than $\hat{\mathbb{P}}_1$ alone. It remains to find a metric measuring this approximation, while controlling the target generalization error (3). To this aim, we assume that the loss functions $\ell_\theta$ belongs, up to a constant, to some RKHS $\mathcal{H}$ (Assumption 4.1).

**Assumption 4.1.** For a kernel $\kappa$, for all $\theta \in \Theta$, $\exists c_\theta \in \mathbb{R}$ and $h_\theta \in \mathcal{H}$ such that $\ell_\theta(z) = c_\theta + h_\theta(z)$ for $z \in \mathcal{Z}$.

The constant $c_\theta$ allows to cover a broader class of loss functions, including constant losses that are not contained in standard RKHSs such as the one induced by the Gaussian kernel.
*Example* 4.2 (Linear regression). The ridge loss $\ell_\theta : z \mapsto (\langle \alpha, x \rangle + \beta - y)^2 + \lambda \|\theta\|^2$, where $\theta = (\alpha, \beta) \in \mathbb{R}^{d+1}$ and $z = (x, y) \in \mathbb{R}^{d+1}$ satisfies Assumption 4.1 for the polynomial kernel $\kappa(z, z') = (\langle z, z' \rangle + 1)^2$ with $c_\theta = \lambda \|\theta\|^2$.

This allows us to control the generalization error in function of the MMD (Eq. 5) of the two distributions $\widehat{\mathbb{P}}(\widehat{\omega})$ and $\mathbb{P}_1$.

**Lemma 4.3.** *Under Assumption 4.1, for any learned weights $\widehat{\omega}$, we have:*

$$\mathbb{E}\Big[ \sup_{\theta \in \Theta} \big| \widehat{\mathcal{R}}_{\widehat{\omega}}(\theta) - \mathcal{R}_1(\theta) \big| \Big] \leq R_\Theta \mathbb{E}\Big[ \mathrm{MMD}_\kappa \big( \mathbb{P}_1, \widehat{\mathbb{P}}(\widehat{\omega}) \big) \Big], \tag{10}$$

*where $R_\Theta = \sup_\theta \|h_\theta\|_{\mathcal{H}}$. Moreover, if for some $r > 0$, $\{h \in \mathcal{H} : \|h\|_{\mathcal{H}} = r\} \subset \{h_\theta\}_{\theta \in \Theta}$, then:*

$$\mathbb{E}\Big[ \sup_{\theta \in \Theta} \big( \widehat{\mathcal{R}}_1(\theta) - \mathcal{R}_1(\theta) \big)^2 \Big] \geq r^2 \frac{\mathrm{Tr}\,\Sigma_1}{n_1}, \tag{11}$$

*where $\Sigma_1 = \Sigma(\mathbb{P}_1)$ is the covariance of $\mathbb{P}_1$ in $\mathcal{H}$.*

This lemma, combined with Eq. (3), shows that the control of the MMD distance between the mixture and the target distribution directly controls the excess risk. The lower bound (11) indicates that to improve upon the naive local estimator, the mixture must achieve an MMD of at most $\sqrt{\mathrm{Tr}\,\Sigma_1/n_1}$; otherwise, training solely on the local empirical distribution is preferable.

**Link with high-dimensional multiple mean estimation.**
As discussed in Section 2.2, the MMD corresponds to the distance between KMEs (Eq. 5). Since the KME of a mixture of distributions is the convex combination of the individual KMEs, controlling the MMD in (10) reduces to:

$$\mathbb{E}\Big[ \mathrm{MMD}_\kappa^2 \big( \mathbb{P}_1, \widehat{\mathbb{P}}(\widehat{\omega}) \big) \Big] = \mathbb{E}\Big[ \Big\| \sum_{k=1}^{B} \widehat{\omega}_k \widehat{\mu}_k - \mu_1 \Big\|_{\mathcal{H}}^2 \Big], \tag{12}$$

where $\widehat{\mu}_k = \widehat{\mu}_{\mathbb{P}_k}$ is the empirical KME of agent $k$. In conclusion, Eq. (12) indicates that finding the weights $\widehat{\omega}$ minimizing the upper-bound in Lemma 4.3 is equivalent to estimating the KME $\mu_1$ with the aggregated empirical KMEs. Those objects being high-dimensional means, we transformed our initial objective to a high-dimensional mean estimation problem with multiple data sources. In the next section, we leverage recent work of Blanchard et al. (2024) that have tackled this problem in general settings.

## 4.2. Learning the Mixture Weights by Q-Aggregation

To estimate the mixture weights $\widehat{\omega}$, we adopt the Q-aggregation method of Blanchard et al. (2024), originally developed for multiple estimation of high-dimensional means, which may in particular correspond to KMEs of different distributions. We emphasize that while this method is not new, the theoretical results that follow are novel as they are derived for the specific KME estimation setting.

The key insight behind this approach is that determining whether two high dimensional means are close to each other is often easier than estimating them precisely (Baraud, 2002;

Blanchard et al., 2018; Blanchard & Fermanian, 2023). In the infinite-dimensional setting, the "high-dimensional effect" is measured through a notion of *effective dimension* $d^e(\mathbb{P})$ of the distribution:

$$d^e(\mathbb{P}) := \frac{\operatorname{Tr}\Sigma(\mathbb{P})}{\|\Sigma(\mathbb{P})\|_{op}}, \; d_1^e := d^e(\mathbb{P}_1), \qquad (13)$$

This notion is also referred to as intrinsic dimension (Hsu et al., 2012; Tropp, 2015) or effective rank (Koltchinskii & Lounici, 2016). For isotropic distributions, it coincides with the actual support dimension, but it remains well-defined in infinite-dimensional settings. Intuitively, $d^e(\mathbb{P})$ quantifies the degrees of freedom of the distribution.

The Q-aggregation method is detailed in Algorithm 1. It relies on an unbiased estimation $\widehat{L}_1(\omega)$ of the mean squared error (in our case, the MMD) of the convex aggregation of empirical means (in our case, the KMEs, see Eq. 12). The weights $\widehat{\omega}$ are then obtained by minimizing this empirical error with a penalization term that accounts for the high-dimensional effect, depending on the (estimated) covariance $\widehat{\Sigma}_1$ of the target distribution. Intuitively, this penalty ensures the error is not underestimated and is controlled uniformly. Algorithm 1 is stated for a general Hilbert space $\mathcal{H}$ and can be implemented whenever scalar products in $\mathcal{H}$ are computable, which holds both in finite dimension and in RKHS. The optimization reduces to quadratic form $\omega^T\widehat{A}\omega + \langle\beta,\omega\rangle$ over the simplex $\mathcal{S}_B$, where $\widehat{A}$ is the Gram matrix of the vectors $\widehat{\nu}_k - \widehat{\nu}_1$. This minimization can be performed, for example, via exponential gradient descent (Kivinen & Warmuth, 1997). For clarity, additional implementation details are deferred to Appendix A.

---

**Algorithm 1** Q-aggregation method

---

**Input:** Hilbert space $(\mathcal{H}, \langle\cdot,\cdot\rangle_{\mathcal{H}})$, empirical means $\widehat{\nu}_k$ of each agent, dataset $(\Phi_i^{(1)})_{i=1}^{n_1}$ of targeted agent 1, bound on data $M$, $C_Q, C_P > 0$

**Let $\widehat{\Sigma}_1$ be the empirical covariance operator:**
$\widehat{\Sigma}_1(\cdot) = \frac{1}{n_1-1}\sum_{i=1}^{n_1}\langle\Phi_i^{(1)} - \widehat{\nu}_1, \cdot\rangle_{\mathcal{H}}(\Phi_i^{(1)} - \widehat{\nu}_1)$

**Compute empirical error:** for $\omega \in \mathcal{S}_B$
$\widehat{L}_1(\omega) := \big\|\sum_{k=1}^B \omega_k\widehat{\nu}_k - \widehat{\nu}_1\big\|_{\mathcal{H}}^2 + 2\omega_1\frac{\operatorname{Tr}\widehat{\Sigma}_1}{n_1}$

**Compute penalization terms:** for $\omega \in \mathcal{S}_B$
$\widehat{Q}_1(\omega) := \frac{1}{\sqrt{n_1}}\sum_{k=2}^B \omega_k\langle\widehat{\nu}_1 - \widehat{\nu}_k, \widehat{\Sigma}_1(\widehat{\nu}_1 - \widehat{\nu}_k)\rangle_{\mathcal{H}}^{1/2}$
$\widehat{P}_1(\omega) := \frac{M}{n_1}\sum_{k=2}^B \omega_k\|\widehat{\nu}_k - \widehat{\nu}_1\|_{\mathcal{H}}$

**Compute weights:**
$\widehat{\omega} \in \arg\min_{\omega\in\mathcal{S}_B}\big(\widehat{L}_1(\omega) + C_Q\widehat{Q}_1(\omega) + C_P\widehat{P}_1(\omega)\big)$

**Output** Return $\widehat{\omega}$

---

Blanchard et al. (2024) showed that this algorithm achieves an optimal trade-off, as restated in Theorem F.1. In the context of KME estimation, their result can be further refined, see Theorem 4.4 below. This refinement is possible because, unlike the general high-dimensional mean estimation setting,

where the mean and covariance of a distribution are separate degrees of freedom, in the KME setting the distance between the covariances of two distributions is controlled by the distance between their respective KMEs (see Lemma F.2).

**Theorem 4.4.** *Let $\mathcal{H}$ be a RKHS with a kernel bounded by $M = 1$, $u_0 := \log(Bn_1)$, and $\widehat{\omega}$ be the output of Algorithm 1 for $\widehat{\nu}_k = \widehat{\mu}_k$ the empirical KMEs, $\Phi_i^{(1)} = \phi_\kappa(Z_i^{(1)})$ the dataset of agent 1 injected in $\mathcal{H}$ and $C_Q^2, C_P > C_0 u_0$ for some absolute constant $C_0$. Then, for any set of agents $V$ that includes agent 1, we have:*

$$\mathbb{E}\Big[\operatorname{MMD}^2\big(\widehat{\mathbb{P}}(\widehat{\omega}), \mathbb{P}_1\big)\Big] \leq \Big[\Delta_V^2 + \frac{\operatorname{Tr}\Sigma_1 + 2\Delta_V}{n_V}\Big]$$
$$+ \frac{Cu_0}{\sqrt{n_1}}\bigg(\sqrt{\frac{|V|-1}{n_V}} \vee \frac{1}{\sqrt{n_1}}\bigg)\bigg(\frac{\operatorname{Tr}\Sigma_1}{\sqrt{d_1^e}} \vee \frac{u_0}{\sqrt{n_1}}\bigg), \quad (14)$$

*where $C > 0$ is some absolute constant depending on $C_Q, C_P$ and $C_0$, and*

$$\Delta_V = \sup_{k\in V}\operatorname{MMD}(\mathbb{P}_1, \mathbb{P}_k), \; and \; n_V = \sum_{k\in V}n_k. \quad (15)$$

**Discussion.** Theorem 4.4 demonstrates the adaptivity of Algorithm 1: the learned mixture achieves a near-optimal bias-variance trade-off, with the bias determined by the distance $\Delta_V$ of the selected distributions to the target $\mathbb{P}_1$, and the variance controlled by the combined sample size $n_V$ of the selected agents. Since the bound (14) holds for any subset $V$, it is in particular valid for the optimal set of agents minimizing it. The ratio $|V|/n_V$ of the selected agents also appears in the bound, reflecting the balance between the number of agents and their data size. Intuitively, the method improves estimation whenever enough agents have distributions close to the target ($\Delta_V$ small) and sufficient data to contribute, yielding a mixture reduces the error below the error $\operatorname{Tr}\Sigma_1/n_1$ (Eq. 6) of the naive local estimate. More generally, the bound is always at least as good as the naive estimator, up to lower-order terms (second part of Eq. 14). These terms are effectively of smaller order in high dimension ($d_1^e$ large) and for a sufficiently large number of local data points, typically $u_0 \simeq \log B \leq \sqrt{n_1}$. Otherwise, the penalization may lead the method to only consider the local data. Below, we illustrate this with an example of agent structure and the resulting performance gains.

*Example* 4.5 (Identical agents). Suppose there a subset $V$ of agents whose distributions are identical to $\mathbb{P}_1$ and who each have at least $n_1$ points, then:

$$\mathbb{E}\Big[\operatorname{MMD}^2\big(\widehat{\mathbb{P}}(\widehat{\omega}), \mathbb{P}_1\big)\Big] \leq \frac{\operatorname{Tr}\Sigma_1}{n_V} + \frac{Cu_0}{n_1}\bigg(\frac{\operatorname{Tr}\Sigma_1}{\sqrt{d_1^e}} \vee \frac{u_0}{\sqrt{n_1}}\bigg).$$

In this case, the aggregation achieves performance comparable to an oracle that directly selects these agents. The improvement is capped by a factor $\sqrt{\min(d_1^e, n_1)}$. Conversely, if $V = \{1\}$, i.e. no other agents have the same

distribution, the first term dominates, recovering the naive estimation error of order $\operatorname{Tr} \Sigma_1 / n_1$.

### 4.3. Controlling the Excess Risk of the Estimator

As outlined above, learning the mixture weights is only the first step of our approach. The second step is to learn the model $\widehat{\theta} \in \arg\min \sum \widehat{\omega}_k \widehat{\mathcal{R}}_k(\theta)$ that minimizes the weighted empirical risk. Corollary 4.6 provides excess risk guarantees for this procedure, assuming $\widehat{\theta}$ is an exact minimizer (i.e. ignoring optimization error).

**Corollary 4.6.** *Under Ass. 4.1, for $\widehat{\theta} \in \arg\min_\theta \widehat{\mathcal{R}}_{\widehat{\omega}}(\theta)$:*

$$\mathbb{E}\big[\mathcal{R}_1(\widehat{\theta})\big] - \mathcal{R}_1(\theta^*) \leq 2R_\Theta \mathbb{E}\big[\mathrm{MMD}\big(\widehat{\mathbb{P}}(\widehat{\omega}), \mathbb{P}_1\big)\big], \quad (16)$$

*where $R_\Theta = \sup_{\theta \in \Theta} \|h_\theta\|_{\mathcal{H}}$. In particular, if $\widehat{\omega}$ are the weights defined in Theorem 4.4 for the RKHS $\mathcal{H}$, the right side of* (16) *is upper bounded by the square root of* (14).

This result shows that minimizing the weighted risk with Q-aggregated weights directly translates into control over the target agent's excess risk, linking the statistical benefit of collaboration to the MMD distance between the aggregated and target distributions.

## 5. Practical Federated Algorithm

In the previous section, we presented a general algorithm for learning the mixture weights of aggregated empirical risks and derived an excess risk bound for its minimizer. However, a closer look at Algorithm 1 and its practical implementation for RKHS and KMEs shows that, for general kernels, all pairwise distances $\kappa(Z_i^{(1)}, Z_j^{(k)})$ between agent 1's data and agent $k$'s data must be computed (details in Appendix A). In practice, this would require centralizing the data (e.g., sharing it with agent 1), which violates the core principles of federated learning. Fortunately, for certain kernels, this is not necessary, as illustrated by the following example.

*Example* 5.1 (Linear regression, continued). Building on Example 4.2, for a second-order polynomial kernel, the empirical KMEs can be transmitted directly, since they depend only on the local empirical mean $\bar{Z}_k = n_k^{-1} \sum_{i=1}^{n_k} Z_i^{(k)}$ and uncentered covariance $\bar{C}_k = n_k^{-1} \sum_{i=1}^{n_k} Z_i^{(k)} (Z_i^{(k)})^T$. Specifically, for $z \in \mathbb{R}^{d+1}$:

$$\widehat{\mu}_k(z) = 1 + 2\langle z, \bar{Z}_k \rangle + z^T \bar{C}_k z. \quad (17)$$

The inner products between KMEs required by Algorithm 1 (for empirical error and penalization terms) can then be computed directly from these quantities:

$$\langle \widehat{\mu}_k, \widehat{\mu}_\ell \rangle_{\mathcal{H}} = 1 + 2\langle \bar{Z}_k, \bar{Z}_\ell \rangle + \operatorname{Tr} \bar{C}_k \bar{C}_\ell. \quad (18)$$

Unfortunately, to the best of our knowledge, similar simplifications do not exist for most popular kernels, such as

Gaussian or Laplace. To address this, we next present a practical algorithm based on random Fourier features.

### 5.1. Random Fourier Features Approximation

As recalled in Section 2.3, Random Fourier Features (RFF) provide a finite-dimensional approximation of the RKHS $\mathcal{H}$ and associated KMEs. Using shared coefficients $\Gamma$, each agent can compute its approximated KME $\widehat{\mu}_k^\Gamma$ locally and transmit it to the server. These KMEs are represented as vectors in $\mathbb{R}^D$, which can then be used in Algorithm 1 to estimate some weights $\widehat{\omega}^\Gamma$. The procedure is given below.

---

**Algorithm 2** Federated Q-aggregation with RFF

**Input:** Distribution $p$ associated to kernel $\kappa$, $D \in \mathbb{N}$, local datasets $Z_\bullet$, $C_Q, C_P > 0$.
**Sampling:** central server samples RFF coefficients: $w_s \sim p, b_s \sim \mathcal{U}([0, 2\pi])$ for $s \in [\![D]\!]$.
**Sharing:** central server shares RFF coefficients $\Gamma = (w_s, b_s)_{s=1}^D$ to all agents.
**Local KME:** each agent $k$ compute $\widehat{\mu}_k^\Gamma = \frac{1}{n_k} \sum_{i=1}^{n_k} \Phi_i^{(k)}$ with $\Phi_i^{(k)} = \big( \cos(\langle w_s, Z_i^{(k)} \rangle + b_s) \big)_{s=1}^D \in \mathbb{R}^D$.
**Sharing KMEs:** $(\widehat{\mu}_k^\Gamma)_{k=2}^B$ are transmitted to agent 1.
**Computing weights $\widehat{\omega}^\Gamma$:** Agent 1 runs Algorithm 1 with $\widehat{\nu}_k = \widehat{\mu}_k^\Gamma$, $(\Phi_i^{(1)})_{i=1}^{n_1}$, $C_Q, C_P$ and $M = \sqrt{2}$.
**Minimizing weighted risk:** run FedAvg to obtain $\widehat{\theta} \in \arg\min_{\theta \in \Theta} \sum_{k=1}^B \widehat{\omega}_k^\Gamma \widehat{\mathcal{R}}_k(\theta)$
**Output:** $\widehat{\theta}$

---

Note that the distribution $p$ is fully determined by the kernel $\kappa$; for instance, $p$ is Gaussian for the Gaussian kernel (see Rahimi & Recht, 2007 for additional examples). By construction, the RFFs are always bounded by a constant, here $M = \sqrt{2}$. Finally, the optimization of the weighted risk in Algorithm 2 is performed using FedAvg, but any other federated optimization method could be used instead.

Theorem 5.2 below gives a bound on the excess risk of the model learned by Algorithm 2. Similarly to Corollary 4.6, it comes from a control of the MMD distance between the resulting empirical measure and the target one.

**Theorem 5.2.** *Let $\kappa$ be a translation-invariant kernel bounded by 1. Let $u_0 = \log B n_1$, and $\hat{\theta}_{RFF}$ the output of Algorithm 2 applied with $C_Q^2, C_P > C_0 u_0$ for some absolute constant $C_0$. Then, under Assumption 4.1, we have:*

$$\mathbb{E}\big[\mathcal{R}_1(\hat{\theta}_{RFF})\big] - \mathcal{R}_1(\theta^*) \leq 2R_\Theta \sqrt{\mathbb{E}\Big[\mathrm{MMD}^2\big(\widehat{\mathbb{P}}(\widehat{\omega}^\Gamma), \mathbb{P}_1\big)\Big]},$$

*where $R_\Theta = \sup_{\theta \in \Theta} \|h_\theta\|_{\mathcal{H}}$ and for any set $V$ with $1 \in V$:*

$$\mathbb{E}\Big[\mathrm{MMD}^2\big(\widehat{\mathbb{P}}(\widehat{\omega}^\Gamma), \mathbb{P}_1\big)\Big] \leq \Big[\Delta_V^2 + \frac{\operatorname{Tr} \Sigma_1}{n_V} + \frac{2\Delta_V}{n_V}\Big]$$

$$+ \frac{C u_0}{\sqrt{n_1}} \Big(\sqrt{\frac{|V|-1}{n_V}} \vee \frac{1}{\sqrt{n_1}}\Big) \Big(\frac{\operatorname{Tr} \Sigma_1}{\sqrt{d_1^e}} \vee \sqrt{\frac{d_1^e}{D}} \vee \frac{u_0}{\sqrt{n_1}}\Big) + C\sqrt{\frac{\log B}{D}}$$

*where $\Sigma_1$ is the covariance operator of $\mathbb{P}_1$ in $\mathcal{H}$, $d_1^e$ its effective dimension and $\Delta_V$, $n_V$ are defined in* (15).

**Discussion.** Compared to Theorem 4.4 and Corollary 4.6, the rate derived here contains additional error terms of order $\mathcal{O}(D^{-\frac{1}{2}})$, which arise from the RFF approximation. It implies that, in theory, $D$ should be chosen larger than $n_1^2$ to expect an improvement. Indeed, the precision of the approximation of the KMEs improves as $D$ increases, and when $D \to \infty$, we recover the result (14). However, larger $D$ increases the dimension of vectors that must be shared, thereby increasing the communication costs. In practice, the choice of $D$ may depend on the federated optimization algorithm used in the last step. Notably, standard `FedAvg` already requires iterative communication of gradients at each round, with dimension equal to the parameter space $\Theta$, so setting $D$ on this scale does not significantly increase communication. Another advantage of our approach is that KMEs $\widehat{\mu}_k^\Gamma$ need to be shared only once, allowing all agents to learn their weights locally in parallel.

### 5.2. Choice of Kernel

Our method has so far been presented in a general setting where the kernel $\kappa$ is defined on an arbitrary data domain $\mathcal{Z}$. The functions in the RKHS associated with a universal kernel (Sriperumbudur et al., 2008; 2010; 2011), such as the Gaussian or Laplace kernel, can approximate any bounded continuous function, which allows us to consider Assumption 4.1 as being approximately satisfied in general (see discussion in Appendix D).

Still, the choice of the kernel and the space on which it operates should be adapted to the application context, the data and the type of heterogeneity, as it determines the information transmitted and compared between agents. In unsupervised learning, many tasks, such as density estimation (maximum likelihood), clustering or dimensionality reduction, can be cast as population risk minimization problems like (1). In such cases, our framework is applicable regardless of the heterogeneity, and using a universal kernel is generally a good option. In supervised learning settings, however, where $Z = (X, Y)$, treating $Z$ as a single random vector, as done in the previous sections, rather than as a pair $(X, Y)$ with different dimensions, may lead to failures or suboptimal performance depending on the type of heterogeneity.

For the case of concept shift, where the conditional distributions $Y|X$ vary across agents, it is important to use a kernel that places sufficient weight on the $Y$ component, especially when $X$ is high-dimensional. In our experiments, we use a weighted Gaussian kernel that emphasizes the $y$ coordinate of $z$, defined as $\kappa_A(z, z') = \exp\left(-z^T A z'\right)$ where $z = (x, y) \in \mathbb{R}^d \times \mathbb{R}$ and $A = \sigma_X I_d \oplus \sigma_Y I_1$. This approach can naturally be generalized to incorporate different forms

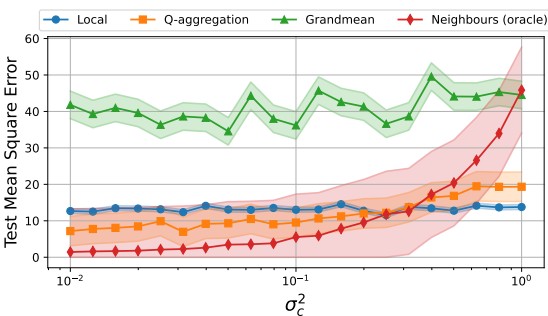

*Figure 1.* Mean Squared Error and its standard deviation of different approaches in function of the intra-group noise $\sigma_c^2$.

of a priori knowledge about the heterogeneity among agents.

In the case of covariate shift, the conditional distribution of $Y|X$ is shared across agents, but the marginal distribution of the features $X$ may vary. Here, we propose to learn the weights from the aggregation of KMEs of the features $X$ alone, rather than the full tuple $(X, Y)$, as the heterogeneity arises from the features. The kernel is defined over $\mathcal{X}$, and so are the RFF. Theoretical guarantees specific to this setting are provided in Appendix C.

## 6. Experiments

We consider different FL settings with heterogeneous agents. We compare Algorithm 2 to *Local* training, i.e. using only local data, and to *GrandMean*, where a single model minimizes the weighted risk $\widehat{\mathcal{R}}_{\boldsymbol{\omega}^{\text{gm}}}$ where $(\omega^{\text{gm}})_k = \frac{n_k}{\sum_{\ell=1}^B n_\ell}$. In some experiments, we further define a notion of *oracle* having a priori information on agents' similarity.

For all experiments, we use the RFF approximation of a Gaussian kernel with $D = 500$ for synthetic experiments and $D = 1000$ for the `FEMNIST` dataset. We emphasize that we *do not* tune the hyperparameters $C_Q$ and $C_P$ of our mixture weight learning approach: they are fixed according to the theory (see experimental details in Appendix B). This illustrates the robustness of our approach to hyperparameter selection.

### 6.1. Synthetic Concept Shift

We consider a concept shift setting in linear regression. The feature distributions of the agents are identical, with $X_i^{(k)} \sim \mathcal{N}\left((1, \ldots, 1), I_d\right)$, whereas the output distributions vary across agents. Each agent belongs to a group $I_k \sim \mathcal{U}(\{-1, 1\})$, with an intra-group proximity determined by a parameter $\sigma_c^2$. For an agent $k \in [\![B]\!]$:

$$Y_i^{(k)} \sim \langle \beta_k, X_i^{(k)} \rangle + \mathcal{N}(0, \sigma_Y^2), \tag{19}$$

$$\beta_k \sim I_k \sqrt{1 - \sigma_c^2}\beta_0 + \sigma_c \varepsilon_k, \quad \beta_0, \varepsilon_k \sim \mathcal{N}(0, I_d). \tag{20}$$

We consider $B = 100$ agents having $n_k = 10$ points each. As $\sigma_c$ increases, agents belonging to the same group become

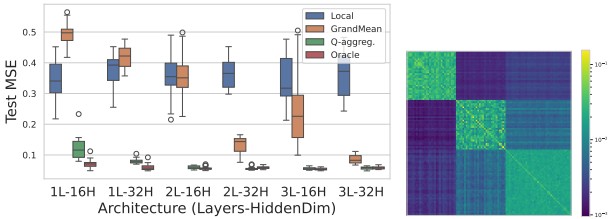

*Figure 2.* Synthetic concept shift. Left side: test MSE in function of the architecture (lower is better). Right side: learned weights.

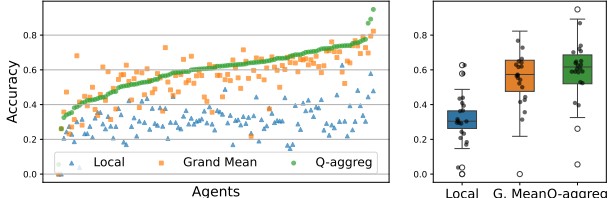

*Figure 3.* FEMNIST. Accuracy of each agent for each method sorted in function of the Q-aggregation ones and a boxplot of these accuracies other the agents (higher is better)

increasingly dispersed: when $\sigma_c = 0$, the parameter $\beta_k$ is identical within each cluster, whereas for $\sigma_c = 1$, the parameters $\beta_k$ are completely independent. The parametrization in Eq. (20) ensures that $\mathbb{E}\left[\|\beta_k\|^2\right] = d$ for all values of $\sigma_c$. Thus, locally, the intrinsic difficulty of the problem remains unchanged. The oracle method learns with the agents of the same group by assigning uniform weights like $\boldsymbol{\omega}^{\mathrm{gm}}$, but without taking $\sigma_c$ into consideration.

In Figure 1, we plot the mean squared error (evaluated using $N_M = 1000$ test points and averaged over $N_r = 100$ different training datasets) of the four baselines, as a function of $\sigma_c^2$. We observe a transition at $\sigma_c^2 = 0.5$. For lower values, agents within the same group are relatively close, and leveraging group-specific data improves learning. Beyond this point, the intra-group variability becomes too large, and collaboration leads to a degradation of performance, as illustrated by the results of the oracle method. Our method correctly captures this behavior, adapts to the level of heterogeneity, improves the learning when it is possible and reduces collaboration when $\sigma_c^2$ becomes too large.

### 6.2. Synthetic Covariate Shift

We consider a synthetic data generation setup designed to model covariate shift: the conditional distribution $Y|X$ is shared across agents, while the distribution of $X$ varies between agents. For every agent, the output variable is defined as

$$Y = \sin(3X_1) + 0.5X_2^2 + 0.1\sum_{i=3}^{d} X_i + \mathcal{N}(0, 0.04),$$

where $X \in \mathbb{R}^d$ is the feature vector. The agents are partitioned into three groups, with group sizes $K_1 = K_2 = 30$ and $K_3 = 40$, yielding a total of $B = 100$ agents, each with $n_k = 20$ samples. Agents belonging to the same group have similar feature distributions; the precise distributions are given in Eq.(25). Our regressor is a ReLU neural network with increasing architecture complexity. Figure 2 reports the MSE of the different learning methods across architectures for an agent of the first group. In this setting, the oracle is trained exclusively on this first group. Under a covariate shift scenario, one might initially expect that training on all data (*GrandMean*) would be optimal. However, for

small models, the network cannot capture the behavior of each subpopulation, making local learning preferable. Our method, in contrast, can identify similar agents and leverage their information effectively. As model capacity increases, learning over all agents becomes increasingly beneficial, and for the largest models, performance approaches that of the oracle. Figure 2 also displays the weight matrix between agents learned by our approach, where the three clusters are well recovered. Lastly, our method achieves performance very close to that of the oracle.

### 6.3. FEMNIST Dataset

We evaluate our approach on the FEMNIST dataset (Caldas et al., 2018), a federated variant of MNIST. Each agent holds handwritten character data (both digits and letters, $|\mathcal{Y}| = 62$) exhibiting different writing styles, and our goal is to train a separate classification model for each agent. Some agents share similar handwriting styles, which can be leveraged to improve learning. We view this setting as a covariate shift problem and use RFFs of the isotropic Gaussian kernel only on the features.

As shown in Fig. 3, our method consistently improves over the GrandMean approach, which is itself generally superior to local training, but may fail for some specific agents. At the opposite, our method has always better always than local training. A possible reason explaining why GrandMean performs well could be the low level of heterogeneity in characters' writing.

## 7. Conclusion

This work introduces an adaptive algorithm for PFL with strong theoretical guarantees. By assuming that the target loss lies in an RKHS, we reformulate the problem as a high-dimensional mean (KME) estimation task, bridging two research areas and enabling the derivation of rigorous statistical results. The use of random Fourier features approximations further allows a controlled trade-off between communication costs and statistical efficiency in FL settings. This yields a principled mechanism to aggregate heterogeneous client information while adapting to a target client, with both theoretical bounds and empirical results supporting the

method's robustness, effectiveness, and interpretability.

Several key questions remain open. In particular, the choice of kernel is central and deserves further investigation. It is also important to quantify the privacy loss associated with sharing KMEs, and to better understand how general loss functions can be approximated within an RKHS. Beyond the setting considered in this work, a natural direction for future research is to extend our framework to scenarios where agents learn their models simultaneously by iteratively aggregating gradients rather than risk measures, enabling even greater adaptability.

## Impact Statement

This paper presents work whose goal is to advance the field of machine learning. There are many potential societal consequences of our work, none of which we feel must be specifically highlighted here.

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

# A. Practical implementation of Q-aggregation (Algorithm 1)

## A.1. Practical computation of the terms

Algorithm 1 is presented in a very general form and involves, in its computations, the empirical covariance operator $\widehat{\Sigma}_1$, which may appear impractical to compute. To clarify the implementation of the algorithm, we detail below the computation of each term. We first provide closed-form expressions as functions of the local data $(\Phi_i^{(1)})_i$ and the empirical means $\widehat{\nu}_k$, and then consider the case where $\mathcal{H}$ is an RKHS, expressing each quantity in terms of the kernel $\kappa$ and the data from each sample. These equations are rewriting of expressions from Blanchard et al. (2024).

**Closed forms.** If it is possible to compute the distance between local points and empirical means (finite dimension or specific kernel such as polynomial ones, see Example 5.1), the following expression can be used.

$$\text{Tr}\,\widehat{\Sigma}_1 = \frac{1}{n_1 - 1} \sum_{i=1}^{n_1} \left\| \Phi_i^{(1)} - \widehat{\nu}_1 \right\|_{\mathcal{H}}^2, \tag{21}$$

$$Q(\boldsymbol{\omega}) = \frac{1}{\sqrt{n_1}} \sum_{k=1}^{B} \omega_k \sqrt{q_k}, \quad \text{where} \quad q_k = \frac{1}{n_1 - 1} \sum_{i=1}^{n_1} \left\langle \Phi_i^{(1)} - \widehat{\nu}_1, \widehat{\nu}_k - \widehat{\nu}_1 \right\rangle_{\mathcal{H}}^2 \tag{22}$$

**General kernel form.** For certain kernels, such as the Gaussian or Laplacian kernel, computing distances between kernel mean embeddings (KMEs) necessitates access to the entire dataset. Consequently, the algorithm is restricted to centralized settings or must rely on random Fourier features, as outlined in Section 5.1, which effectively reduces the problem to the previously discussed formulations.

$$\text{Tr}\,\widehat{\Sigma}_1 = \frac{1}{2(n_1 - 1)} \sum_{i \neq j=1}^{n_1} \left( \kappa(Z_i^{(1)}, Z_i^{(1)}) - 2\kappa(Z_i^{(1)}, Z_j^{(1)}) + \kappa(Z_j^{(1)}, Z_j^{(1)}) \right), \tag{23}$$

and

$$q_k = \frac{1}{n_1 - 1} \sum_{i=1}^{n_1} \left( \frac{1}{n_k} \sum_{j=1}^{n_k} \kappa(Z_i^{(1)}, Z_j^{(k)}) - \frac{1}{n_1} \sum_{j=1}^{n_1} \kappa(Z_i^{(1)}, Z_j^{(1)}) \right)^2$$
$$- \frac{n_1}{n_1 - 1} \left( \frac{1}{n_1 n_k} \sum_{i=1}^{n_1} \sum_{j=1}^{n_k} \kappa(Z_i^{(1)}, Z_j^{(k)}) - \frac{1}{n_1^2} \sum_{i=1}^{n_1} \sum_{j=1}^{n_1} \kappa(Z_i^{(1)}, Z_j^{(1)}) \right)^2 \tag{24}$$

## A.2. Optimization

The minimized quantity of Algorithm 1 can be expressed as a quadratic form. Indeed for any $\boldsymbol{\omega} \in \mathcal{S}_B$:

$$\widehat{L}_1(\boldsymbol{\omega}) + C_Q \widehat{Q}_1(\boldsymbol{\omega}) + C_P \widehat{P}_1(\boldsymbol{\omega}) = \boldsymbol{\omega}^T A \boldsymbol{\omega} + \langle \boldsymbol{\omega}, b \rangle,$$

where $A = \left( \langle \widehat{\nu}_k - \widehat{\nu}_1, \widehat{\nu}_\ell - \widehat{\nu}_1 \rangle_{\mathcal{H}} \right)_{k,\ell=1}^{T}$ and $b$ is just the vectorial sum of $\widehat{P}$ and $\widehat{Q}$ and adding $\text{Tr}\,\widehat{\Sigma}_1/n_1$ at the first-coordinate . We then find the minimum of this quadratic form by an exponential gradient descent (Kivinen & Warmuth, 1997). To a better a convergence, we adjust it using the Prox-Method of Nemirovski (2004). The learning rate is chosen as $\eta = c / \left( 2\|A\|_{op} + \|b\|_\infty \right)$, since $2\|A\|_{op} + \|b\|_\infty$ upper bounds the Lipschitz norm of the gradient of $\boldsymbol{\omega} \mapsto \boldsymbol{\omega}^T A \boldsymbol{\omega} + \langle b, \boldsymbol{\omega} \rangle$. The parameter $c$ is fixed at $c = 0.5$ and the number of gradients step at $T = 1000$ in all the experiments.

---

**Algorithm 3** Exponential gradient descent

**Inputs.** Gradient $\nabla f : \mathbb{R}^B \to \mathbb{R}^B$, initialization point $\omega_0 \in \mathcal{S}_B$, learning rate $\eta > 0$, number of steps T.
**for** $t = 0, \ldots, T - 1$ **do**
    **Compute gradient.** $g_t = \nabla f(\omega_t)$
    **Proxy update.** For $i \in [\![B]\!]$, $(\omega_t')_i = (\omega_t)_i \dfrac{e^{-\eta(g_t)_i}}{\sum_{j=1}^{B} e^{-\eta(g_t)_j}}$
    **Compute proxy gradient.** $g_t' = \nabla f(\omega_t')$
    **Update.** For $i \in [\![B]\!]$, $(\omega_{t+1})_i = (\omega_t)_i \dfrac{e^{-\eta(g_t')_i}}{\sum_{j=1}^{B} e^{-\eta(g_t')_j}}$
**end for**
**Output:** $\omega_T$.

---

# B. Technical details of the experiments

For experiments involving neural networks, we report the performance of each method corresponding to the best test accuracy achieved during training. This choice avoids the need to tune stopping times for the different algorithms, which is itself a nontrivial issue in the federated learning setting.

## B.1. Synthetic concept shift

Table 1 presents the different parameter used in the experiments of Section 6.1. To capture the concept shift, we rescaled the features impact on the kernel.

| Type | Parameter | Value |
|---|---|---|
| Data | Dimension | $d = 20$ |
| | Noise variance | $\sigma_Y^2 = 2$ |
| | Number of points per agent | $n_k = 10$ |
| | Number of agents | $B = 100$ |
| | Number of repetitions | $N_r = 100$ |
| | Number of test points for evaluating MSE | $N_M = 1000$ |
| Method | Dimension of random features | $D = 500$ |
| | Kernel | $\kappa((x, y), (x', y')) = \exp\left(-\|x - x'\|^2 / \sqrt{d+1} - (y - y')^2\right)$ |
| | RFF distribution | $p \sim \mathcal{N}(0, A)$ with $A = \begin{bmatrix} \frac{I_d}{d+1} & 0 \\ 0 & 1 \end{bmatrix}$ |
| | Parameter of Q-aggregation | $C_Q^2 = C_P = \log B$ |
| | Model | Linear regression |

Table 1. Parameters of the synthetic concept shift experiments of Section 6.1

## B.2. Synthetic covariate shift

Table 2 presents the different parameter used in the experiments of Section 6.2.

| Type | Parameter | Value |
|---|---|---|
| Data | Dimension | $d = 4$ |
| | Intra group variance | $v_1^2 = 0.01, v_2^2 = 0.3$ |
| | Variance | $\sigma_1^2 = 0.4, \sigma_2^2 = 0.8$ |
| | Number of points per agent | $n_k = 20$ |
| | Number of agents by group | $K_1 = K_2 = 30$ |
| | Center of group 2 | $\mu_0 = (2, \ldots, 2)$ |
| | Number of agents by group | $B = 100$ |
| | Number of repetitions | $N_r = 20$ |
| | Number of test points for evaluating MSE | $N_M = 2000$ |
| Method | Dimension of random features | $D = 500$ |
| | Kernel | Gaussian kernel |
| | RFF distribution | $p \sim \mathcal{N}(0, I_d)$ |
| | Parameter of Q-aggregation | $C_Q^2 = C_P = \log B$ |
| | Model | ReLU neural networks |
| | Architecture | Number of hidden layers $\in \{1, 2, 3\}$, hidden dimension $\in \{16, 32\}$ |
| | Number of epochs | $n_e = 2000$ |
| | Learning rate | $lr = 0.001$ |

Table 2. Parameters of the synthetic covariance shift experiments of Section 6.2

The distribution of the features is

$$X_i^{(k)} \sim \begin{cases} \mathcal{N}(\mu_k, \sigma_1^2 I_d), \ 1 \le k \le K_1, \ \mu_k \sim \mathcal{N}(0, v_1^2 I_d), \\ \mathcal{N}(\mu_k, \sigma_2^2 I_d), \ K_1 < k \le K_1 + K_2, \ \mu_k \sim \mathcal{N}(\mu_0, v_2^2 I_d), \\ \mathcal{U}([-6,6]^d), \ K_1 + K_2 < k \le B. \end{cases} \tag{25}$$

### B.3. Femnist dataset

The model used in the experiments of Section 6.3 in a ReLU neural network with 1 hidden layer of dimension 32. It is trained during 2000 epochs with a learning rate of 0.001. We only consider $B = 192$ agents of this dataset. The test and train sizes are represented in Figure 4. The RFFs are the Gaussian ones with a dimension $D = 1000$. The ambient dimension of the features is $d = 28 \times 28 = 782$. The Q-aggregation is applied with $C_Q^2 = C_P = \log B$.

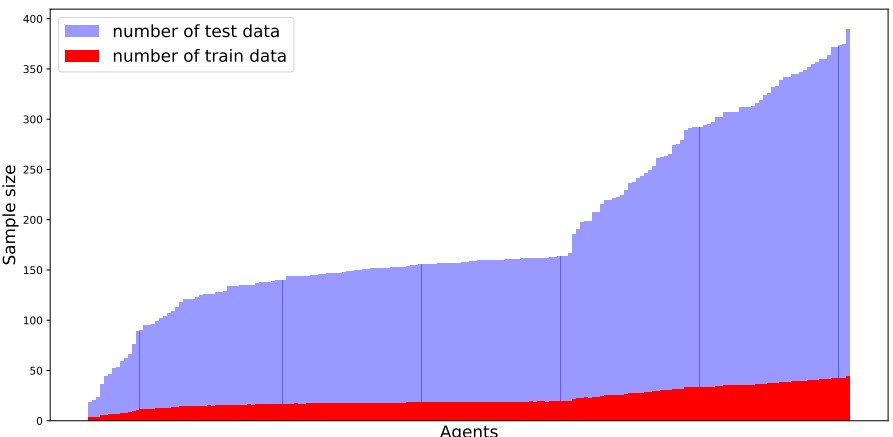

*Figure 4.* Number of train and test points for each agent.

## C. Theoretical result in case of covariate shift

We present in this section some theoretical results for our approach in case of covariate shift. In this case, we propose to directly learned the aggregation weights from the KME of the features instead of the KMEs of the tuple. Then, under Assumption C.1, the approximation error of the risk is controlled by the MMD distance between the (weighted) empirical distribution of the features and the one of the target distribution, and does not involve the dependence in $Y$.

**Assumption C.1** (Covariate shift). $\mathcal{Z} = \mathcal{X} \times \mathcal{Y}$ and for a kernel $\kappa$ on $\mathcal{X}$, for all $\theta \in \Theta$ and $y \in \mathcal{Y}$, there exists $c_\theta \in \Theta$ and $h_{\theta,y} \in \mathcal{H}_\mathcal{X}$ such that $\ell_\theta(x, y) = c_\theta + h_{\theta,y}(x)$.

**Proposition C.2.** *Let Assumption C.1 be satisfied and that the conditional distribution $\mathbb{P}^{Y|X}$ is common over the agents, i.e. $\mathbb{P}_k = \mathbb{P}^{Y|X}\mathbb{P}_k^X$ for $k \in [\![B]\!]$. Then, for any $\widehat{\omega}$ the weights and $\theta \in \Theta$:*

$$\left| \mathbb{E}\left[ \widehat{\mathcal{R}}_{\widehat{\omega}}(\theta) \right] - \mathcal{R}_1(\theta) \right| \le R_\theta^{\mathcal{Y}} \mathbb{E}\left[ \mathrm{MMD}\left( \mathbb{P}_1^X, \widehat{\mathbb{P}}^X(\widehat{\omega}) \right) \right] \tag{26}$$

*where $\widehat{\mathbb{P}}^X(\omega) = \sum_{k=1}^B \omega_k \widehat{\mathbb{P}}_k^X$ is the empirical mixture of the features and $R_\theta^{\mathcal{Y}} = \sup_{y \in \mathcal{Y}} \|h_{\theta,y}\|_{\mathcal{H}_\mathcal{X}}$.*

This result justifies the choice of a kernel just defined on the features, the aggregation would then lead to an optimal aggregation of the features distribution of the agents. However, we emphasize that the guarantee (26) is weaker than those used in the rest of the paper such as (10) or (16), since the control of the risk approximation holds only for a fixed $\theta$ and not uniformly over $\Theta$.

**Proof of Proposition C.2.** Without loss of generality we can assume $c_\theta = 0$. Let $H_\theta \in \mathcal{H}_X$ be defined by $H_\theta(x) = \mathbb{E}[h_{\theta,Y}(x)|X = x]$, then for any $\theta \in \Theta$:

$$\mathcal{R}_1(\theta) = \mathbb{E}[H_\theta(X)] = \left\langle H_\theta, \mu_{\mathbb{P}_1^X} \right\rangle, \tag{27}$$

where $\mathbb{P}_1^X$ is the distribution of the features of $\mathbb{P}_1$. Moreover:

$$\mathbb{E}\Big[\widehat{\mathcal{R}}_{\widehat{\boldsymbol{\omega}}}(\theta)\Big] = \mathbb{E}\Bigg[\sum_{k=1}^{B}\widehat{\omega}_k \frac{1}{n_k}\sum_{i=1}^{n_k}\mathbb{E}\Big[h_{\theta,Y_i^{(k)}}(X_i^{(k)})|X_i^{(k)}\Big]\Bigg] = \mathbb{E}\Bigg[\sum_{k=1}^{B}\widehat{\omega}_k \frac{1}{n_k}\sum_{i=1}^{n_k}H_\theta(X_i^{(k)})\Bigg] = \mathbb{E}\Bigg[\Big\langle H_\theta, \sum_{k=1}^{B}\widehat{\omega}_k\widehat{\mu}_{\mathbb{P}_k^X}\Big\rangle_{\mathcal{H}_X}\Bigg], \tag{28}$$

where $\widehat{\mu}_{\mathbb{P}_k^X}$ is the KME of the empirical distribution of the features of agent $k$. We have used for the conditioning that the weights $\widehat{\omega}$ are learned from the features $X_i^{(k)}$. Combining (27) and (28) leads to the result:

$$\Big|\mathbb{E}\Big[\widehat{\mathcal{R}}_{\widehat{\boldsymbol{\omega}}}(\theta)\Big] - \mathcal{R}_1(\theta)\Big| \leq \Bigg|\mathbb{E}\Bigg[\Big\langle H_\theta, \mu_{\mathbb{P}_1^X} - \sum_{k=1}^{B}\widehat{\omega}_k\widehat{\mu}_{\mathbb{P}_k^X}\Big\rangle_{\mathcal{H}_X}\Bigg]\Bigg| \leq R_\theta^{\mathcal{Y}}\mathbb{E}\Big[\mathrm{MMD}\big(\mathbb{P}_1^X, \widehat{\mathbb{P}}^X(\widehat{\boldsymbol{\omega}})\big)\Big], \tag{29}$$

using that $\|H_\theta\|_{\mathcal{H}_X} \leq \mathbb{E}\Big[\|h_{\theta,Y}\|_{\mathcal{H}_X}|X=x\Big] \leq R_\theta^{\mathcal{Y}}$, by Jensen's inequality. $\qquad\square$

## D. Universal kernel and approximation of the loss function

An important property of kernels is universality. A kernel defined on a space $\mathcal{Z}$ is said to be universal if its associated RKHS $\mathcal{H}$ is dense in the space of continuous functions with respect to the uniform norm. That is, for any continuous function $f$ on $\mathcal{Z}$ and any $\varepsilon > 0$, there exists a function $h \in \mathcal{H}$ in the RKHS such that its distance to $f$ in the uniform norm is smaller than $\varepsilon$, $\|f - h\|_\infty < \varepsilon$. Many kernels have been identified for different types of spaces with this property; see for example Muandet et al. (2014) for a review.

Our approach relies on the assumption that the loss functions $\ell_\theta$ belong to the RKHS, up to a constant term $c_\theta$ (Assumption 4.1). For a universal kernel, this assumption is expected to hold approximately: for any loss function, there exists a function $h_{\theta,\varepsilon}$ in the RKHS at distance at most $\varepsilon$. However, in order to obtain a bound on the excess risk, we must also control the RKHS norm of such a function, which, to the best of our knowledge, is not generally quantifiable. This issue is formalized in the following lemma.

**Lemma D.1.** *Let $\widehat{\theta} \in \arg\min_{\theta \in \Theta} \widehat{\mathcal{R}}_{\widehat{\boldsymbol{\omega}}}(\theta)$ for some weights $\widehat{\boldsymbol{\omega}} \in \mathcal{S}_B$, then:*

$$\mathbb{E}\Big[\mathcal{R}_1(\widehat{\theta})\Big] - \mathcal{R}_1(\theta_1^*) \leq \min_{\varepsilon \geq 0}\Big[2R_{\Theta,\varepsilon}\mathbb{E}\Big[\mathrm{MMD}\Big(\widehat{\mathbb{P}}_{\widehat{\boldsymbol{\omega}}}, \mathbb{P}_1\Big)\Big] + 4\varepsilon\Big] \tag{30}$$

*where:*

$$R_{\Theta,\varepsilon} = \sup_{\theta \in \Theta} \min_{\substack{c > 0, h \in \mathcal{H}: \\ \|h+c-\ell_\theta\|_\infty < \varepsilon}} \|h\|_{\mathcal{H}}. \tag{31}$$

If Assumption 4.1 is satisfied, we recover Lemma 4.3. As a RKHS contained generally regular functions, the quantity $R_{\Theta,\varepsilon}$ can be interpreted as the complexity of the model relatively to the kernel.

**Proof of Lemma D.1.** Let $\varepsilon \geq 0$ such that $R_{\Theta,\varepsilon}$ is finite. Then for any $\ell_\theta$ there exists $h_\theta \in \mathcal{H}$ and $c_\theta > 0$ such that $\|h+c-\ell_\theta\|_\infty < \varepsilon$. Then:

$$\mathbb{E}\Big[\mathcal{R}_1(\widehat{\theta})\Big] = \mathbb{E}\big[\ell_{\widehat{\theta}}(Z)\big] \leq \varepsilon + c_\theta + \mathbb{E}\big[h_{\widehat{\theta}}(Z)\big]. \tag{32}$$

As $h_{\widehat{\theta}} \in \mathcal{H}$, then $\mathbb{E}\big[h_{\widehat{\theta}}(Z)\big] - \sum_{k=1}^{B}\widehat{\omega}_k\frac{1}{n_k}\sum_{i=1}^{n_k}h_{\widehat{\theta}}(Z_i^{(k)}) \leq R_{\Theta,\varepsilon}\mathbb{E}\Big[\mathrm{MMD}\Big(\widehat{\mathbb{P}}_{\widehat{\boldsymbol{\omega}}}, \mathbb{P}_1\Big)\Big]$. Using again the proximity with $\ell_\theta$ and the definition of $\widehat{\theta}$, we obtain:

$$\sum_{k=1}^{B}\widehat{\omega}_k\frac{1}{n_k}\sum_{i=1}^{n_k}h_{\widehat{\theta}}(Z_i^{(k)}) \leq \sum_{k=1}^{B}\widehat{\omega}_k\frac{1}{n_k}\sum_{i=1}^{n_k}\ell_{\widehat{\theta}}(Z_i^{(k)}) - c_\theta + \varepsilon \leq \sum_{k=1}^{B}\widehat{\omega}_k\frac{1}{n_k}\sum_{i=1}^{n_k}\ell_{\theta^*}(Z_i^{(k)}) - c_\theta + \varepsilon$$
$$= \widehat{\mathcal{R}}_{\widehat{\boldsymbol{\omega}}}(\theta^*) - c_\theta + \varepsilon.$$

Using the same transformation as at the beginning of the proof, but with $\theta^*$ in place of $\widehat{\theta}$, we obtain the final result. $\qquad\square$

# E. Random Fourier Features results

### E.1. Approximation

We begin this section by a control in probability between the MMD in the original RKHS and the MMD computed in the random fourier features RKHS.

**Lemma E.1.** *Let $\mathbb{P}$ and $\mathbb{Q}$ two distributions on $\mathcal{Z}$, $\mu_{\mathbb{P}}$ and $\mu_{\mathbb{Q}}$ their respective KME in $\mathcal{H}$ and $\mu_{\mathbb{P}}^{\Gamma}$ and $\mu_{\mathbb{Q}}^{\Gamma}$ their respective KME in the random Fourier features RKHS $\mathcal{H}_{\Gamma}$. For any $\delta \in (0, 1)$:*

$$\mathbb{P}_{\Gamma}\left[\|\mu_{\mathbb{P}} - \mu_{\mathbb{Q}}\|_{\mathcal{H}} \leq \|\mu_{\mathbb{P}}^{\Gamma} - \mu_{\mathbb{Q}}^{\Gamma}\|_{\mathcal{H}_{\Gamma}} + C\sqrt{\frac{\log \delta^{-1}}{D}}\right] \geq 1 - \delta, \tag{33}$$

*for some constant $C > 0$ depending only on a bound on the kernel.*

*Proof.* For the rest of the proof $X, X'$ (resp. $Y, Y'$) will denote independent random variables of distribution $\mathbb{P}$ (resp. $\mathbb{Q}$) and $\phi_{\Gamma}(z) = (D^{-1/2}\phi(\gamma_i, z))$ will be the random Fourier features map. We recall that:

$$\|\mu_{\mathbb{P}} - \mu_{\mathbb{Q}}\|_{\mathcal{H}}^2 = \mathbb{E}[\kappa(X, X')] - 2\mathbb{E}[\kappa(X, Y)] + \mathbb{E}[\kappa(Y, Y')], \tag{34}$$

and:

$$\|\mu_{\mathbb{P}}^{\Gamma} - \mu_{\mathbb{Q}}^{\Gamma}\|_{\mathcal{H}_{\Gamma}}^2 = \frac{1}{D}\sum_{i=1}^{D}\left(\mathbb{E}[\phi(\gamma_i, X)|\gamma_i] - \mathbb{E}[\phi(\gamma_i, Y)|\gamma_i]\right)^2.$$

Let us first compute the expectation over the random Fourier features $\Gamma$ of the KME distance in $\mathcal{H}_{\Gamma}$. As the features $\gamma_i$ are i.i.d., we get

$$\begin{aligned}
\mathbb{E}_{\gamma_1}\left[\|\mu_{\mathbb{P}}^{\Gamma} - \mu_{\mathbb{Q}}^{\Gamma}\|_{\mathcal{H}_{\Gamma}}^2\right] &= \mathbb{E}_{\Gamma}\left[\left(\mathbb{E}[\phi(\gamma_1, X)|\gamma_1] - \mathbb{E}[\phi(\gamma_1, Y)|\gamma_1]\right)^2\right] \\
&= \mathbb{E}_{\gamma_1}\left[\mathbb{E}[\phi(\gamma_1, X)\phi(\gamma_1, X')|\gamma_1] - 2\mathbb{E}[\phi(\gamma_1, X)\phi(\gamma_1, Y)|\gamma_1] + \mathbb{E}[\phi(\gamma_1, Y)\phi(\gamma_1, Y')|\gamma_1]\right] \\
&= \mathbb{E}_{\gamma_1}\left[\phi(\gamma_1, X)\phi(\gamma_1, X') - 2\phi(\gamma_1, X)\phi(\gamma_1, Y) + \phi(\gamma_1, Y)\phi(\gamma_1, Y')\right],
\end{aligned}$$

after developing the square. Using that for any $z, z' \in \mathcal{Z}$, $\mathbb{E}_{\gamma_1}[\phi(\gamma_1, z)\phi(\gamma_1, z')] = \kappa(z, z')$, we obtain:

$$\mathbb{E}_{\gamma_1}\left[\|\mu_{\mathbb{P}}^{\Gamma} - \mu_{\mathbb{Q}}^{\Gamma}\|_{\mathcal{H}_{\Gamma}}^2\right] = \mathbb{E}[\kappa(X, X')] - 2\mathbb{E}[\kappa(X, Y)] + \mathbb{E}[\kappa(Y, Y')] = \|\mu_{\mathbb{P}} - \mu_{\mathbb{Q}}\|_{\mathcal{H}}^2.$$

Applying Bernstein's inequality to the i.i.d. random variables $Z_i = \left(\mathbb{E}[\phi(\gamma_i, X)|\gamma_i] - \mathbb{E}[\phi(\gamma_i, Y)|\gamma_i]\right)^2 - \|\mu_{\mathbb{P}} - \mu_{\mathbb{Q}}\|_{\mathcal{H}}^2$, we get that, with probability at least $1 - \delta$:

$$\|\mu_{\mathbb{P}} - \mu_{\mathbb{Q}}\|_{\mathcal{H}}^2 \leq \|\mu_{\mathbb{P}}^{\Gamma} - \mu_{\mathbb{Q}}^{\Gamma}\|_{\mathcal{H}_{\Gamma}}^2 + C\sqrt{\frac{\log \delta^{-1}}{D}}\|\mu_{\mathbb{P}} - \mu_{\mathbb{Q}}\|_{\mathcal{H}} + C\frac{\log \delta^{-1}}{D}, \tag{35}$$

for some absolute constant $C$. We have used that the random variables $Z_i$ are upper bounded by a constant as $\phi$ and the kernel $\kappa$ are supposed bounded and that $\text{Var}[Z_i] \leq \mathbb{E}[Z_i^2] \leq C\mathbb{E}[Z_i]$. After inverting (35), we get

$$\|\mu_{\mathbb{P}} - \mu_{\mathbb{Q}}\|_{\mathcal{H}} \leq \|\mu_{\mathbb{P}}^{\Gamma} - \mu_{\mathbb{Q}}^{\Gamma}\|_{\mathcal{H}_{\Gamma}} + C\sqrt{\frac{\log \delta^{-1}}{D}}, \tag{36}$$

which concludes the proof. $\square$

**Lemma E.2.** *Let $\mathbb{Q}, \mathbb{P}_1, \ldots, \mathbb{P}_B$ some distributions on $\mathcal{Z}$, $\mu_{\mathbb{Q}}^{\Gamma}$, $\mu_{\mathbb{P}_k}^{\Gamma}$ their respective KME in the random Fourier features space $\mathcal{H}_{\Gamma}$ of dimension $D$ for the kernel $\kappa$. Let $\boldsymbol{\omega}^{\Gamma}$ some weights in the Simplex $\mathcal{S}_B$ depending on the features $\Gamma$. Then:*

$$\mathbb{E}_{\Gamma}\left[\text{MMD}_{\kappa}^2\left(\sum_{k=1}^{B}\omega_k^{\Gamma}\mathbb{P}_k, \mathbb{Q}\right)\right] \leq \mathbb{E}_{\Gamma}\left[\left\|\sum_{k=1}^{B}\omega_k^{\Gamma}\mu_{\mathbb{P}_k}^{\Gamma} - \mu_{\mathbb{Q}}^{\Gamma}\right\|_{\mathcal{H}_{\Gamma}}^2\right] + C\sqrt{\frac{\log B}{D}}. \tag{37}$$

*Proof.* We reuse the notations of the proof of Lemma E.1. Let us first remark that with probability at least $1 - \delta$, for any distributions $\mathbb{P}, \mathbb{Q}$:

$$\langle \mu_{\mathbb{Q}}^{\Gamma}, \mu_{\mathbb{P}}^{\Gamma} \rangle_{\mathcal{H}_{\Gamma}} = \frac{1}{D} \sum_{i=1}^{D} \mathbb{E}_{X \sim \mathbb{P}, Y \sim \mathbb{Q}}[\phi(\gamma_i, X)\phi(\gamma_i, Y)] \geq \mathbb{E}_{X \sim \mathbb{Q}, Y \sim \mathbb{Q}}[\kappa(X, Y)] - C\sqrt{\frac{\log \delta^{-1}}{D}}. \tag{38}$$

This is obtained by remarking that $\mathbb{E}_{\Gamma}[\mathbb{E}_{X \sim \mathbb{P}, Y \sim \mathbb{Q}}[\phi(\gamma_i, X)\phi(\gamma_i, Y)]] = \mathbb{E}_{X \sim \mathbb{P}, Y \sim \mathbb{Q}}[\kappa(X, Y)] = \langle \mu_{\mathbb{P}}, \mu_{\mathbb{Q}} \rangle$ and are upper bounded by construction of the random Fourier features (see Section 2.3) and by applying Hoeffding's inequality. It follows, after an union bound over all pair of distributions, with probability at least $1 - \delta$:

$$\left\| \sum_{k=1}^{B} \omega_k^{\Gamma} \mu_{\mathbb{P}_k}^{\Gamma} - \mu_{\mathbb{Q}}^{\Gamma} \right\|_{\mathcal{H}_{\Gamma}}^2 = \sum_{k,\ell=1}^{B} \omega_k^{\Gamma} \omega_\ell^{\Gamma} \langle \mu_{\mathbb{P}_k}^{\Gamma} - \mu_{\mathbb{Q}}^{\Gamma}, \mu_{\mathbb{P}_\ell}^{\Gamma} - \mu_{\mathbb{Q}}^{\Gamma} \rangle_{\mathcal{H}_{\Gamma}}$$

$$= \sum_{k,\ell=1}^{B} \omega_k^{\Gamma} \omega_\ell^{\Gamma} \left( \langle \mu_{\mathbb{P}_k}^{\Gamma}, \mu_{\mathbb{P}_\ell}^{\Gamma} \rangle_{\mathcal{H}_{\Gamma}} - \langle \mu_{\mathbb{P}_k}^{\Gamma}, \mu_{\mathbb{Q}}^{\Gamma} \rangle_{\mathcal{H}_{\Gamma}} - \langle \mu_{\mathbb{P}_\ell}^{\Gamma}, \mu_{\mathbb{Q}}^{\Gamma} \rangle_{\mathcal{H}_{\Gamma}} + \langle \mu_{\mathbb{Q}}^{\Gamma}, \mu_{\mathbb{Q}}^{\Gamma} \rangle_{\mathcal{H}_{\Gamma}} \right)$$

$$\geq \sum_{k,\ell=1}^{B} \omega_k^{\Gamma} \omega_\ell^{\Gamma} \left( \langle \mu_{\mathbb{P}_k}, \mu_{\mathbb{P}_\ell} \rangle_{\mathcal{H}} - \langle \mu_{\mathbb{P}_k}, \mu_{\mathbb{Q}} \rangle_{\mathcal{H}} - \langle \mu_{\mathbb{P}_\ell}, \mu_{\mathbb{Q}} \rangle_{\mathcal{H}} + \langle \mu_{\mathbb{Q}}, \mu_{\mathbb{Q}} \rangle_{\mathcal{H}} \right) - C\sqrt{\frac{\log((B+1)\delta^{-1})}{D}}$$

$$= \mathrm{MMD}_{\kappa}^2 \left( \sum_{k=1}^{B} \omega_k^{\Gamma} \mathbb{P}_k, \mathbb{Q} \right) - C\sqrt{\frac{\log((B+1)\delta^{-1})}{D}}.$$

As the inequality is satisfied with probability $\delta$ for any $\delta > 0$, we can integrate and obtain that for some constant $C$:

$$\mathbb{E}_{\Gamma} \left[ \mathrm{MMD}_{\kappa}^2 \left( \sum_{k=1}^{B} \omega_k^{\Gamma} \mathbb{P}_k, \mathbb{Q} \right) \right] \leq \mathbb{E}_{\Gamma} \left[ \left\| \sum_{k=1}^{B} \omega_k^{\Gamma} \mu_{\mathbb{P}_k}^{\Gamma} - \mu_{\mathbb{Q}}^{\Gamma} \right\|_{\mathcal{H}_{\Gamma}}^2 \right] + C\sqrt{\frac{\log B}{D}}.$$

A careful reader can notice that this inequality is an equality when $B = 1$, which allows to upper bound $\log(B + 1)$ by $C \log B$ for $B \geq 2$ and some absolute constant $C > 0$. $\qquad \square$

### E.2. Covariance operators

**Definition E.3** (Covariance operator). Let $\mathbb{Q}$ a squared integrable distribution in an Hilbert space $\mathcal{H}$. The covariance operator of $\mathbb{Q}$ is defined as:

$$\begin{cases} \mathcal{H} & \to \mathcal{H} \\ v & \mapsto \mathbb{E}_{\Phi \sim \mathbb{Q}}[\langle v, \Phi \rangle_{\mathcal{H}} \Phi] - \langle \mathbb{E}_{\Phi \sim \mathbb{Q}}[\Phi], v \rangle_{\mathcal{H}} \mathbb{E}_{\Phi \sim \mathbb{Q}}[\Phi] = \mathbb{E}_{\Phi \sim \mathbb{Q}}[\langle v, \Phi - \mathbb{E}_{\Phi \sim \mathbb{Q}}[\Phi] \rangle_{\mathcal{H}} \Phi] \end{cases} \tag{39}$$

By abuse of language, we refer to the **covariance of a distribution** $\mathbb{P}$ **on** $\mathcal{Z}$ as the covariance operator of the pushforward $\phi_\kappa(\mathbb{P})$ of $\mathbb{P}$ into the RKHS $\mathcal{H}$, that is, of the distribution $\mathbb{Q}$ on $\mathcal{H}$ where $\Phi = \phi_\kappa(Z) \sim \mathbb{Q}$.

For sake of clarity, we recall briefly below the definition of the trace and operator norm of a Hilbert-Schmidt operator. which aligns with the definition in finite dimension.

**Definition E.4** (Trace and operator norm). Let $\Sigma : \mathcal{H} \to \mathcal{H}$ be a trace class operator over a separable Hilbert space $(\mathcal{H}, \langle \cdot, \cdot \rangle_{\mathcal{H}})$. Then:

$$\|\Sigma\|_{op} := \sup_{h: \|h\|_{\mathcal{H}} = 1} \|\Sigma h\|_{\mathcal{H}}, \quad \text{and} \quad \mathrm{Tr}\, \Sigma := \sum_{k=1}^{\infty} \langle e_k, \Sigma e_k \rangle_{\mathcal{H}}, \tag{40}$$

where $(e_k)_k$ is a countable orthonormal basis.

**Lemma E.5.** *Let* $\Sigma, S$ *the respective covariance operators of centered distribution* $\mathbb{P}$ *and* $\mathbb{Q}$. *Then* $\mathrm{Tr}\, \Sigma^2 := \mathbb{E}_{Z, Z' \sim \mathbb{P}}\left[ \langle Z, Z' \rangle^2 \right]$ *and* $\mathrm{Tr}\, \Sigma S = \mathbb{E}_{Z \sim \mathbb{P}, Z' \sim \mathbb{Q}}\left[ \langle Z, Z' \rangle^2 \right]$.

*Proof.* Let us prove the two statements simultaneously. As $\Sigma$ and $S$ are Hilbert Schmidt operators, for $(e_k)_{k \geq 1}$ an orthonormal basis of $\mathcal{H}$:

$$\operatorname{Tr} \Sigma S = \sum_{k=1}^{\infty} \langle e_k, \Sigma S e_k \rangle_{\mathcal{H}} = \sum_{k=1}^{\infty} \langle \Sigma e_k, S e_k \rangle_{\mathcal{H}} = \mathbb{E}_{Z \sim \mathbb{P}, Z' \sim \mathbb{Q}} \left[ \sum_{k=1}^{\infty} \langle Z, e_k \rangle \langle Z', e_k \rangle \langle Z, Z' \rangle \right], \tag{41}$$

using the linearity of the expectation. Then using again Parseval's identity, we get:

$$\operatorname{Tr} \Sigma S = \mathbb{E}_{Z \sim \mathbb{P}, Z' \sim \mathbb{Q}} \left[ \langle Z, Z' \rangle^2 \right].$$

Taking $S = \Sigma$ gives the second statement. $\qquad\square$

**Lemma E.6.** *Trace of covariance operator of RFF Let $\Sigma^{\Gamma}$ the covariance operator of the random Fourier features $\phi^{\Gamma}(Z) \in \mathcal{H}_{\Gamma}$ for $Z \sim \mathbb{P}$. Then*

$$\mathbb{E}_{\Gamma} \left[ \operatorname{Tr} \Sigma^{\Gamma} \right] = \operatorname{Tr} \Sigma, \tag{42}$$

*where $\Sigma$ is the covariance operator of the mapping $\phi_{\kappa}(Z) = \kappa(Z, \cdot)$ in the original RKHS $\mathcal{H}$.*

*Proof.* Let us just use that for any covariance operator $\Sigma$ of a distribution $\mathbb{P}$, $\operatorname{Tr} \Sigma = \mathbb{E}_{Z, Z' \sim \mathbb{P}} \left[ \|Z - Z'\|^2 \right] / 2$. It follows:

$$
\begin{aligned}
2 \mathbb{E}_{\Gamma} \left[ \operatorname{Tr} \Sigma^{\Gamma} \right] &= \mathbb{E}_{\Gamma} \left[ \mathbb{E}_{Z, Z' \sim \mathbb{P}} \left[ \left\| \phi^{\Gamma}(Z) - \phi^{\Gamma}(Z') \right\|_{\mathcal{H}_{\Gamma}}^2 \right] \right] \\
&= \mathbb{E}_{Z, Z' \sim \mathbb{P}} \left[ \mathbb{E}_{\Gamma} \left[ \left\| \phi^{\Gamma}(Z) \right\|_{\mathcal{H}_{\Gamma}}^2 - 2 \langle \phi^{\Gamma}(Z), \phi^{\Gamma}(Z') \rangle_{\mathcal{H}_{\Gamma}} + \left\| \phi^{\Gamma}(Z') \right\|_{\mathcal{H}_{\Gamma}}^2 \right] \right] \\
&= \mathbb{E}_{Z, Z' \sim \mathbb{P}} [\kappa(Z, Z) - 2\kappa(Z, Z') + \kappa(Z', Z')] = 2 \mathbb{E}_{Z, Z' \sim \mathbb{P}} \left[ \|\kappa(Z, \cdot) - \kappa(Z', \cdot)\|_{\mathcal{H}}^2 \right] = 2 \operatorname{Tr} \Sigma,
\end{aligned}
$$

which concludes the proof. $\qquad\square$

**Lemma E.7** (Expectation of operator norm). *Let $\phi_1, \dots, \phi_D$ i.i.d. centered random vectors in an Hilbert space $\mathcal{H}$. Let*

$$\widehat{\Sigma}(\cdot) = \frac{1}{D} \sum_{i=1}^{D} \langle \cdot, \phi_i \rangle_{\mathcal{H}} \phi_i, \tag{43}$$

*be the empirical covariance operator. Then:*

$$\|\Sigma\|_{op} \leq \mathbb{E} \left[ \left\| \widehat{\Sigma} \right\|_{op} \right] \leq \|\Sigma\|_{op} + \sqrt{\frac{\mathbb{E}_{\phi \sim \mathbb{P}} \left[ \|\phi\|_{\mathcal{H}}^4 \right]}{D}}$$

*Proof.* Using triangle inequality, we have the following inequality:

$$\left\| \widehat{\Sigma}_{op} \right\| \leq \|\Sigma\|_{op} + \left\| \widehat{\Sigma} - \Sigma \right\|_{op} \leq \|\Sigma\|_{op} + \sqrt{\operatorname{Tr} \left( \widehat{\Sigma} - \Sigma \right)^2}. \tag{44}$$

Let us now bound the trace of the square of the operators. Let $\widehat{\mathbb{P}} = \frac{1}{D} \sum_{i=1}^{D} \delta_{\phi_i}$ the empirical distribution. Then $\widehat{\Sigma}$ is the covariance operator of $\widehat{\mathbb{P}}$. It follows using Lemma E.5:

$$
\begin{aligned}
\operatorname{Tr} \left( (\widehat{\Sigma} - \Sigma)^2 \right) &= \operatorname{Tr} \left( \widehat{\Sigma}^2 - \widehat{\Sigma} \Sigma - \Sigma \widehat{\Sigma} + \Sigma^2 \right) = \mathbb{E}_{\phi, \phi' \sim \widehat{\mathbb{P}}} \left[ \langle \phi, \phi' \rangle_{\mathcal{H}}^2 \right] - 2 \mathbb{E}_{\phi \sim \mathbb{P}, \phi' \sim \widehat{\mathbb{P}}} \left[ \langle \phi, \phi' \rangle_{\mathcal{H}}^2 \right] + \operatorname{Tr} \Sigma^2 \\
&= \frac{1}{D^2} \sum_{i,j=1}^{D} \langle \phi_i, \phi_j \rangle_{\mathcal{H}}^2 - \frac{2}{D} \sum_{i=1}^{D} \mathbb{E}_{\phi} \left[ \langle \phi_i, \phi \rangle^2 \right]_{\mathcal{H}} + \operatorname{Tr} \Sigma^2.
\end{aligned}
$$

Let us now take the expectation over the $\phi_i$-s. We get:

$$\mathbb{E} \left[ \operatorname{Tr}(\widehat{\Sigma} - \Sigma)^2 \right] = \frac{1}{D} \mathbb{E}_{\phi \sim \mathbb{P}} \left[ \|\phi\|_{\mathcal{H}}^4 \right] + \operatorname{Tr} \Sigma^2 \left( \frac{D(D-1)}{D^2} - 2 + 1 \right) \leq \frac{\mathbb{E}_{\phi \sim \mathbb{P}} \left[ \|\phi\|_{\mathcal{H}}^4 \right]}{D}$$

We conclude by Jensen's inequality. $\qquad\square$

**Lemma E.8.** *Let $G$ the covariance of $\phi_\Gamma(Z) \in \mathbb{R}^D$ defined by $G = \mathbb{E}_Z[(\phi_\Gamma(Z) - \mu)^T(\phi_\Gamma(Z) - \mu)]$ with $\mu = \mu(\Gamma) = \mathbb{E}_Z[\phi_\Gamma(Z)]$. Then:*

$$\|\Sigma\|_{op} \leq \mathbb{E}_\Gamma\left[\|G\|_{op}\right] \leq \|\Sigma\|_{op} + \sqrt{\frac{\operatorname{Tr}\Sigma}{D}}, \tag{45}$$

*where $\Sigma : L^2(\mathbb{P}) \to L^2(\mathbb{P})$ is the covariance operator of $\phi(Z) = k(Z, \cdot) \in \mathcal{H}$.*

*Proof.* Let us first remark that $G$ is a Gram matrix of vectors of $L^2(\mathbb{P})$. Let us first denote $\phi_\Gamma(z) = (\phi(\gamma_i, Z))_{i=1}^D$, $\mu_i = \mathbb{E}_{Z \sim \mathbb{P}}[\phi(\gamma_i, z)]$ and $\bar{\phi}_i(\cdot) = \bar{\phi}(\gamma_i, \cdot) - \mu_i$. Then for any $i, j \in \llbracket D \rrbracket$:

$$G_{ij} = \mathbb{E}_Z[(\phi(\gamma_i, Z) - m_i)(\phi(\gamma_j, Z) - m_j)] := \mathbb{E}_Z[\bar{\phi}_i(Z)\bar{\phi}_j(Z)] = \langle \bar{\phi}_i, \bar{\phi}_j \rangle_{L^2(\mathbb{P})}$$

Then:

$$\|G\|_{op} = \sup_{\|u\|_D = 1} u^T G u = \sup_{\|u\|_D = 1} \left\| \sum_{i=1}^D u_i \bar{\phi}_i \right\|_{L^2(\mathbb{P})}^2 = \sup_{\|u\|_D = 1} \sup_{\|v\|_{L^2(\mathbb{P})} = 1} \left\langle v, \sum_{i=1}^D u_i \bar{\phi}_i \right\rangle_{L^2(\mathbb{P})}^2,$$

where the second supremum is taken over the unit ball of $L^2(\mathbb{P})$. We can invert the supremums and use the symmetry:

$$\|G\|_{op} = \sup_{\|v\|_{L^2(\mathbb{P})} = 1} \sup_{\|u\|_D = 1} \left( \sum_{i=1}^D u_i \langle v, \bar{\phi}_i \rangle_{L^2(\mathbb{P})} \right)^2 = \sup_{\|v\|_{L^2(\mathbb{P})} = 1} \left( \sup_{\|u\|_D = 1} \sum_{i=1}^D u_i \langle v, \bar{\phi}_i \rangle_{L^2(\mathbb{P})} \right)^2$$

$$= \sup_{\|v\|_{L^2(\mathbb{P})} = 1} \sum_{i=1}^D \langle v, \bar{\phi}_i \rangle_{L^2(\mathbb{P})}^2 = \sup_{\|v\|_{L^2(\mathbb{P})} = 1} \left\langle v, \widehat{\Sigma} v \right\rangle_{L^2(\mathbb{P})} = \|\widehat{\Sigma}\|_{op},$$

where $\widehat{\Sigma}$ is the empirical covariance operator defined by:

$$\widehat{\Sigma} : \begin{cases} L^2(\mathbb{P}) & \to L^2(\mathbb{P}) \\ v & \mapsto \frac{1}{D}\sum_{i=1}^D \sqrt{D}\bar{\phi}_i \left\langle v, \sqrt{D}\bar{\phi}_i \right\rangle_{L^2(\mathbb{P})}. \end{cases}$$

Effectively, we have that for any $v \in \mathcal{H}$ and $z \in \mathcal{Z}$:

$$\begin{aligned}
\mathbb{E}_\Gamma\left[\widehat{\Sigma}v(z)\right] &= D\mathbb{E}_{\Gamma,Z}[\phi_1(z)v(Z)(\phi_1(Z) - \mu_1)] \\
&= \mathbb{E}_Z[v(Z)\mathbb{E}_\Gamma[D\phi_1(z)\phi_1(Z)]] - \mathbb{E}_{Z,Z'}[v(Z)\mathbb{E}_\Gamma[D\phi_1(z)\phi_1(Z')]] \\
&= \mathbb{E}_Z[v(Z)\kappa(Z, z)] - \mathbb{E}_{Z,Z'}[v(Z)\kappa(z, Z')] \\
&= \mathbb{E}_Z[\langle \kappa(Z, \cdot), v \rangle_\mathcal{H} \kappa(Z, z)] - \langle v, \mu_\mathbb{P} \rangle \mu_\mathbb{P}(z),
\end{aligned}$$

where we recognize the covariance operator. We then use Lemma E.7:

$$\mathbb{E}\left[\|G\|_{op}\right] = \mathbb{E}\left[\|\widehat{\Sigma}\|_{op}\right] \leq \mathbb{E}\left[\|\Sigma\|_{op}\right] + \sqrt{\frac{\mathbb{E}_\Gamma\left[\|\sqrt{D}\bar{\phi}_1\|_{L^2(\mathbb{P})}^4\right]}{D}}.$$

We conclude using that $\sqrt{D}\bar{\phi}_1$ is upper bounded by $1$ and that $\mathbb{E}_\Gamma\left[\|\sqrt{D}\bar{\phi}_1\|_{L^2(\mathbb{P})}^2\right] = \operatorname{Tr}\Sigma$.

*Remark* E.9. We have used that the spectrum of the covariance operator $\Sigma$ in $L^2(\mathbb{P})$ and $\mathcal{H}$ are the same (Rosasco et al., 2010, Proposition 8).

$\square$

# F. Proofs

This section provides detailed proofs for the theoretical results outlined in the main paper.

### F.1. Proof of Example 4.2

Let us recall that $\ell_{(\alpha,\beta)}((x,y)) = (\langle x, \alpha\rangle + \beta - y)^2$. If $\beta \neq 0$, then $\ell_{(\alpha,\beta)}(\cdot) = \beta^2 \kappa((\alpha/\beta, -1/\beta), \cdot)$. For $\beta = 0$:

$$\ell_{(\alpha,\beta)}((x,y)) = (\langle x, \alpha\rangle - y)^2 = \frac{1}{2}\kappa((\alpha, -1), (x,y)) + \frac{1}{2}\kappa((-\alpha, 1), (x,y)) - \kappa(0, (x,y))$$

$\square$

### F.2. Proof of Lemma 4.3

**Upper bound.** As $\ell_\theta(\cdot) = c_\theta + h_\theta(\cdot)$ by assumption, we get, for any $\theta \in \Theta$:

$$\sum_{k=1}^{B} \widehat{\omega}_k \widehat{\mathcal{R}}_k(\theta) - \mathcal{R}_1(\theta) = \sum_{k=1}^{B} \widehat{\omega}_k \mathbb{E}_{Z\sim\widehat{\mathbb{P}}_k}[h_\theta(Z)] - \mathbb{E}_{Z\sim\mathbb{P}_1}[h_\theta(Z)] = \mathbb{E}_{Z\sim\widehat{\mathbb{P}}(\widehat{\omega})}[h_\theta(Z)] - \mathbb{E}_{Z\sim\mathbb{P}_1}[h_\theta(Z)],$$

where $\widehat{\mathbb{P}}(\widehat{\omega}) = \sum_{k=1}^{B} \widehat{\omega}_k \widehat{\mathbb{P}}_k$. It follows, by definition of the MMD (Eq. (5)):

$$\sum_{k=1}^{B} \widehat{\omega}_k \widehat{\mathcal{R}}_k(\theta) - \mathcal{R}_1(\theta) \leq \|h_\theta\|_{\mathcal{H}} \mathrm{MMD}\left(\widehat{\mathbb{P}}(\widehat{\omega}), \mathbb{P}_1\right),$$

which leads to (10).

**Lower bound.** Similarly as above, for any $\theta \in \Theta$:

$$\widehat{\mathcal{R}}_1(\theta) - \mathcal{R}_1(\theta) = \mathbb{E}_{Z\sim\widehat{\mathbb{P}}_1}[h_\theta(Z)] - \mathbb{E}_{Z\sim\mathbb{P}_1}[h_\theta(Z)] = \langle\, h_\theta, \widehat{\mu}_1 - \mu_1\,\rangle_{\mathcal{H}}.$$

As $\{h \in \mathcal{H} : \|h\|_{\mathcal{H}} = r\} \subset \{h_\theta\}_{\theta\in\Theta}$, there exists $\theta \in \Theta$ such that $h_\theta = r\frac{\widehat{\mu}_1 - \mu_1}{\|\widehat{\mu}_1 - \mu_1\|_{\mathcal{H}}}$. It follows:

$$\mathbb{E}\left[\sup_{\theta\in\Theta}\left(\widehat{\mathcal{R}}_1(\theta) - \mathcal{R}_1(\theta)\right)^2\right] \geq r^2 \mathbb{E}\left[\|\widehat{\mu}_1 - \mu_1\|_{\mathcal{H}}^2\right] = r^2\frac{\mathrm{Tr}\,\Sigma_1}{n_1},$$

which concludes the proof. $\square$

### F.3. Proof of Theorem 4.4

We begin by restated a general result of Blanchard et al. (2024) on the Q-aggregation method (Algorithm 1). It is adapted from Eq. (90) p.62 of the proof of Theorem 3 of this work.

**Theorem F.1** (Blanchard et al., 2024, restated). *Let $u_0 \geq 2\log BN_1$, $\widehat{\omega}$ be the output of Algorithm 1 for $\{Z_i^{(1)}\}_{i=1}^{n_1}$ a sample of $\mathbb{P}_1$, $C_Q^2, C_p \geq C_0 u_0$, $\widehat{\mu}_k = n_k^{-1}\sum_{i=1}^{n_k} Z_i^{(k)}$ the empirical means of i.i.d. samples of distribution $\mathbb{P}_k$. Assume that all the distributions are bounded by $M$, then*

$$\mathbb{E}\left[\left\|\sum_{k=1}^{B} \widehat{\omega}_k \widehat{\mu}_k - \mu_1\right\|_{\mathcal{H}}^2\right] \leq \min_{\boldsymbol{\omega}\in\mathcal{S}_B}\left[R_1(\boldsymbol{\omega}) + C\sqrt{u_0}Q_1(\boldsymbol{\omega}) + \frac{CMu_0}{n_1}\sum_{k=2}^{B}\omega_k\left(\|\mu_k - \mu_1\| + \sqrt{\frac{\mathrm{Tr}\,\Sigma_k}{n_k}}\right)\right]$$

$$+ C\frac{\sqrt{\mathrm{Tr}\,\Sigma_1^2}}{n_1}u_0 + CM\frac{\sqrt{\mathrm{Tr}\,\Sigma_1}}{N_1^{3/2}}u_0 + C\frac{M^2}{N_1^2}u_0^2, \tag{46}$$

*where $C > 0$ is an absolute constant depending on $C_0$, $\Sigma_k$ is the covariance of $\mathbb{P}_k$, $n_k$ the sample size of sample $k$ and*

$$R_1(\boldsymbol{\omega}) = \left\|\sum_{k=1}^{B}\omega_k(\mu_k - \mu_1)\right\|_{\mathcal{H}}^2 + \sum_{k=1}^{B}\omega_k^2\frac{\mathrm{Tr}\,\Sigma_k}{n_k}, \tag{47}$$

$$Q(\boldsymbol{\omega}) = \frac{1}{\sqrt{n_1}}\sum_{k=2}^{B}\omega_k\sqrt{\langle\mu_1 - \mu_k, \Sigma_1(\mu_1 - \mu_k)\rangle_{\mathcal{H}} + \frac{\mathrm{Tr}\,\Sigma_1\Sigma_k}{n_k}} \tag{48}$$

The objective is to follow the proof of Theorem 3 of Blanchard et al. (2024) and adapt it using the context that the estimated quantities are KMEs, by using following Lemma F.2 to simplify the bound and the assumptions of the Q-aggregation method.

**Lemma F.2.** *Let $\kappa$ a kernel constant over the diagonal ($\kappa(x,x) = M^2, \forall x$). Let $\mathbb{P}$ and $\mathbb{Q}$ two distributions and $\Sigma_{\mathbb{P}}$ and $\Sigma_{\mathbb{Q}}$ their respective covariance operators in the RKHS. Then:*

$$|\operatorname{Tr}\Sigma_{\mathbb{P}} - \operatorname{Tr}\Sigma_{\mathbb{Q}}| \leq 2M\|\mu_{\mathbb{P}} - \mu_{\mathbb{Q}}\|_{\mathcal{H}}. \tag{49}$$

*Proof.* Let us first remark that:

$$\operatorname{Tr}\Sigma_{\mathbb{P}} = \mathbb{E}_{X\sim\mathbb{P}}\left[\|X - \mu_{\mathbb{P}}\|^2_{\mathcal{H}}\right] = \mathbb{E}[\kappa(X,X)] - \|\mu_{\mathbb{P}}\|^2_{\mathcal{H}} = M^2 - \|\mu_{\mathbb{P}}\|^2_{\mathcal{H}}.$$

Then

$$|\operatorname{Tr}\Sigma_{\mathbb{P}} - \operatorname{Tr}\Sigma_{\mathbb{Q}}| \leq \left|\|\mu_{\mathbb{P}}\|^2_{\mathcal{H}} - \|\mu_{\mathbb{Q}}\|^2_{\mathcal{H}}\right| = \left|\langle\mu_{\mathbb{P}}, \mu_{\mathbb{P}} - \mu_{\mathbb{Q}}\rangle_{\mathcal{H}} + \langle\mu_{\mathbb{P}} - \mu_{\mathbb{Q}}, \mu_{\mathbb{Q}}\rangle_{\mathcal{H}}\right|$$
$$\leq (\|\mu_{\mathbb{P}}\|_{\mathcal{H}} + \|\mu_{\mathbb{Q}}\|_{\mathcal{H}})\|\mu_{\mathbb{Q}} - \mu_{\mathbb{Q}}\|_{\mathcal{H}}.$$

We conclude using that the norms of the KMEs are bounded by M:

$$\|\mu_{\mathbb{P}}\|^2_{\mathcal{H}} = \mathbb{E}[\langle\kappa(X,\cdot), \kappa(X',\cdot)\rangle_{\mathcal{H}}] \leq \mathbb{E}[\|\kappa(X,\cdot)\|_{\mathcal{H}}\|\kappa(X',\cdot)\|_{\mathcal{H}}] = \mathbb{E}\left[\sqrt{\kappa(X,X)\kappa(X',X')}\right] \leq M^2.$$

$\square$

**Proof of Theorem 4.4**  According to Theorem F.1, we have for any $\boldsymbol{\omega} \in \mathcal{S}_B$:

$$\mathbb{E}\left[\|\widehat{\mu}_{\widehat{\boldsymbol{\omega}}} - \mu_1\|^2_{\mathcal{H}}\right] \leq R_1(\boldsymbol{\omega}) + C\sqrt{u_0}Q(\boldsymbol{\omega}) + \frac{Cu_0}{n_1}\sum_{k=2}^{B}\omega_k\left(\|\mu_k - \mu_1\|_{\mathcal{H}} + \sqrt{\frac{\operatorname{Tr}\Sigma_k}{n_k}}\right)$$
$$+ C\frac{\sqrt{\operatorname{Tr}\Sigma_1^2}}{n_1}u_0 + C\frac{\sqrt{\operatorname{Tr}\Sigma_1}}{n_1^{3/2}}u_0 + C\frac{u_0^2}{n_1^2}, \tag{50}$$

where $R_1$ and $Q_1$ are defined respectively in (47) and (48). By remarking that:

$$Q(\boldsymbol{\omega}) \leq \sqrt{\frac{\|\Sigma_1\|_{op}}{n_1}}\sum_{k=2}^{B}\omega_k\left(\|\mu_k - \mu_1\|_{\mathcal{H}} + \sqrt{\operatorname{Tr}\Sigma_k/n_k}\right), \tag{51}$$

it remains to choose a weight $\omega$ to bound effectively this quantity and $R_1(\boldsymbol{\omega})$.

Let $V$ a subset of agents. We first fix $\omega_k = 0$ for $k \notin V$. Then using Lemma F.2:

$$R_1(\boldsymbol{\omega}) \leq (1 - \omega_1)^2\Delta^2 + \omega_1^2\frac{\operatorname{Tr}\Sigma_1}{n_1} + \sum_{k\in V, k\neq 1}\frac{\omega_k^2}{n_k}(\operatorname{Tr}\Sigma_1 + 2\Delta).$$

where $\Delta = \Delta_V = \max_{k\in V}\|\mu_1 - \mu_k\|_{\mathcal{H}}$. Let us then choose:

$$\omega_1 = \frac{\Delta^2 + \frac{\operatorname{Tr}\Sigma_1 + 2\Delta}{n_V - n_1}}{\Delta^2 + \frac{\operatorname{Tr}\Sigma_1 + 2\Delta}{n_V - n_1} + \frac{\operatorname{Tr}\Sigma_1}{n_1}}, \quad \omega_k = (1 - \omega_1)\frac{n_k}{n_V - n_1},$$

Then:

$$R_1(\boldsymbol{\omega}) \leq (1 - \omega_1)^2\left[\Delta^2 + \frac{\operatorname{Tr}\Sigma_1 + 2\Delta}{n_V - n_1}\right] + \omega_1^2\frac{\operatorname{Tr}\Sigma_1}{n_1}$$
$$= \frac{\operatorname{Tr}\Sigma_1}{n_1}\frac{\Delta^2 + \frac{\operatorname{Tr}\Sigma_1 + 2\Delta}{n_V - n_1}}{\Delta^2 + \frac{\operatorname{Tr}\Sigma_1 + 2\Delta}{n_V - n_1} + \frac{\operatorname{Tr}\Sigma_1}{n_1}}$$
$$\leq \Delta^2 + \frac{\operatorname{Tr}\Sigma_1}{n_1}\frac{\frac{\operatorname{Tr}\Sigma_1 + 2\Delta}{n_V - n_1}}{\frac{\operatorname{Tr}\Sigma_1 n_V}{n_1(n_V - n_1)}}$$
$$\leq \Delta^2 + \frac{\operatorname{Tr}\Sigma_1}{n_V} + \frac{2\Delta}{n_V}.$$

For the same weights $\boldsymbol{\omega}$ we now bound (51). Firstly

$$\sum_{k=2}^{B} \omega_k \|\mu_k - \mu_1\|_{\mathcal{H}} \leq \Delta(1 - \omega_1) \leq \Delta \frac{\frac{\operatorname{Tr}\Sigma_1}{n_1}}{\Delta^2 + 0 + \frac{\operatorname{Tr}\Sigma_1}{n_1}} \leq \min\left(\Delta, \sqrt{\frac{\operatorname{Tr}\Sigma_1}{n_1}}\right).$$

For the second part we get:

$$\sum_{k=2}^{B} \omega_k \sqrt{\frac{\operatorname{Tr}\Sigma_k}{n_k}} \leq \sqrt{\operatorname{Tr}\Sigma_1 + 2\Delta} \sum_{k=2}^{B} \frac{\omega_k}{\sqrt{n_k}} = \sqrt{\operatorname{Tr}\Sigma_1 + 2\Delta}(1 - \omega_1) \frac{\sum_{k\in V, k\neq 1} \sqrt{n_k}}{n_V - n_1}.$$

Let us first remark that:

$$\sqrt{\operatorname{Tr}\Sigma_1 + 2\Delta} \leq \sqrt{\operatorname{Tr}\Sigma_1} + \frac{\Delta}{\sqrt{\operatorname{Tr}\Sigma_1}} \tag{52}$$

and that, by concavity

$$\sum_{k\in V, k\neq 1} \sqrt{n_k} \leq \sqrt{(|V| - 1)(n_V - n_1)}.$$

As $(1 - \omega_1) \leq \frac{n_V - n_1}{n_V}$, we get:

$$\sqrt{\operatorname{Tr}\Sigma_1}(1 - \omega_1) \frac{\sqrt{(|V| - 1)(n_V - n_1)}}{n_V - n_1} \leq \sqrt{\operatorname{Tr}\Sigma_1} \sqrt{\frac{(|V| - 1)}{n_V}}.$$

Using that

$$(1 - \omega_1) \frac{\Delta}{\sqrt{\operatorname{Tr}\Sigma_1}} = \frac{\sqrt{\operatorname{Tr}\Sigma_1}}{n_1} \frac{\Delta}{\Delta^2 + \frac{2\Delta}{n_V - n_1} + \frac{\operatorname{Tr}\Sigma_1 n_V}{n_1(n_V - n_1)}} \leq \frac{\sqrt{\operatorname{Tr}\Sigma_1}}{n_1} \sqrt{\frac{n_1(n_V - n_1)}{\operatorname{Tr}\Sigma_1 n_V}} = \sqrt{\frac{n_V - n_1}{n_1 n_V}}$$

and again the concavity, we obtain:

$$\sum_{k=2}^{B} \omega_k \sqrt{\frac{\operatorname{Tr}\Sigma_k}{n_k}} \leq \sqrt{\frac{|V| - 1}{n_V}} \left[\sqrt{\operatorname{Tr}\Sigma_1} + \frac{1}{\sqrt{n_1}}\right]. \tag{53}$$

We can now plug all the bound into (50):

$$\mathbb{E}\left[\|\widehat{\mu}_{\widehat{\omega}} - \mu_1\|_{\mathcal{H}}^2\right] \leq \left[\Delta^2 + \frac{\operatorname{Tr}\Sigma_1}{n_V} + \frac{2\Delta}{n_V}\right] + C\left[\sqrt{\frac{\|\Sigma_1\|_{op} u_0}{n_1}} + \frac{u_0}{n_1}\right]\sqrt{\frac{\operatorname{Tr}\Sigma_1}{n_1}} \tag{54}$$

$$+ C\left[\sqrt{\frac{\|\Sigma_1\|_{op} u_0}{n_1}} + \frac{u_0}{n_1}\right]\sqrt{\frac{|V| - 1}{n_V}}\left[\sqrt{\operatorname{Tr}\Sigma_1} + \frac{1}{\sqrt{n_1}}\right]$$

$$+ C\frac{\sqrt{\|\Sigma_1\|_{op} \operatorname{Tr}\Sigma_1}}{n_1} u_0 + C\frac{\sqrt{\operatorname{Tr}\Sigma_1}}{n_1^{3/2}} u_0 + C\frac{u_0^2}{n_1^2}. \tag{55}$$

After combining the terms and upper bounding $\|\Sigma_1\|_{op} \leq \operatorname{Tr}\Sigma_1 \leq 1$ we get:

$$\mathbb{E}\left[\|\widehat{\mu}_{\widehat{\omega}} - \mu_1\|_{\mathcal{H}}^2\right] \leq \left[\Delta^2 + \frac{\operatorname{Tr}\Sigma_1}{n_V} + \frac{2\Delta}{n_V}\right] + C\frac{\sqrt{\|\Sigma_1\|_{op} \operatorname{Tr}\Sigma_1}}{n_1} u_0 + C\frac{u_0}{n_1^{3/2}} + C\frac{u_0^2}{n_1^2}$$

$$\leq \left[\Delta^2 + \frac{\operatorname{Tr}\Sigma_1}{n_V} + \frac{2\Delta}{n_V}\right] + \frac{Cu_0}{\sqrt{n_1}} \max\left(\sqrt{\frac{|V| - 1}{n_V}}, \frac{1}{\sqrt{n_1}}\right) \max\left(\frac{\operatorname{Tr}\Sigma_1}{\sqrt{d_1^e}}, \frac{u_0}{\sqrt{n_1}}\right).$$

$$\square$$

## F.4. Proof of Corollary 4.6

Assume $\widehat{\theta} \in \arg\min_{\theta \in \Theta} \sum_{k=1}^{B} \widehat{\omega}_k \frac{1}{n_k} \sum_{i=1}^{n_k} \ell_\theta(Z_i^{(k)})$, we neglect the optimization error. Then:

$$\mathbb{E}\big[\mathcal{R}_{\widehat{\theta}}^{(1)}\big] = \mathbb{E}\big[\langle \ell_{\widehat{\theta}}, \mu_1 \rangle_{\mathcal{H}}\big] = \mathbb{E}\big[\langle \ell_{\widehat{\theta}}, \mu_1 - \widehat{\mu} \rangle_{\mathcal{H}} + \langle \ell_{\widehat{\theta}}, \widehat{\mu} \rangle_{\mathcal{H}}\big],$$

where $\widehat{\mu} = \sum_{k=1}^{B} \widehat{\omega}_k \widehat{\mu}_k$ is the convex aggregation of the empirical KMEs of each agent. As for any $\theta$, $\langle \ell_\theta, \widehat{\mu} \rangle_{\mathcal{H}} = \sum_{k=1}^{B} \widehat{\omega}_k \frac{1}{n_k} \sum_{i=1}^{n_k} \ell_\theta(Z_i^{(k)}) = \widehat{\mathcal{R}}_{\widehat{\omega}}(\theta)$, we get by definition of $\widehat{\theta}$ that $\langle \ell_{\widehat{\theta}}, \widehat{\mu} \rangle_{\mathcal{H}} \le \langle \ell_{\theta^*}, \widehat{\mu} \rangle_{\mathcal{H}}$. Using again that $\mathcal{R}_{\theta^*}^{(1)} = \langle \ell_{\theta^*}, \mu_1 \rangle_{\mathcal{H}}$, it follows:

$$\begin{aligned}
\mathbb{E}\big[\mathcal{R}_{\widehat{\theta}}^{(1)}\big] - \mathcal{R}_{\theta^*}^{(1)} &\le \mathbb{E}\big[\langle \ell_{\widehat{\theta}}, \mu_1 - \widehat{\mu} \rangle_{\mathcal{H}} + \langle \ell_{\theta^*}, \widehat{\mu} - \mu_1 \rangle_{\mathcal{H}}\big] \\
&\le \mathbb{E}\big[(\|\ell_{\widehat{\theta}}\|_{\mathcal{H}} + \|\ell_{\theta^*}\|_{\mathcal{H}})\|\widehat{\mu} - \mu_1\|_{\mathcal{H}}\big] \\
&\le 2 \sup_{\theta \in \Theta} \|\ell_\theta\|_{\mathcal{H}} \mathbb{E}[\|\widehat{\mu} - \mu_1\|_{\mathcal{H}}],
\end{aligned}$$

thanks to Cauchy-Schwartz inequality. $\qquad\square$

## F.5. Proof of Theorem 5.2

By combining Lemma 4.3 and Equation (3), we know that controlling the MMD distance between the empirical mixture to $\mathbb{P}_1$ leads to a control of the excess risk. Let us control this quantity.

From Lemma E.2, applied conditionally to the datasets $\mathcal{D}$ to $\mathbb{P}_k \leftarrow \widehat{\mathbb{P}}_k$ and $\mathbb{Q} \leftarrow \mathbb{P}_1$, we first have that:

$$\mathbb{E}_{\Gamma,\mathcal{D}}\Big[\mathrm{MMD}_\kappa^2\Big(\sum_{k=1}^{B} \omega_k^\Gamma \widehat{\mathbb{P}}_k, \mathbb{P}_1\Big)\Big] \le \mathbb{E}_{\Gamma,\mathcal{D}}\Big[\Big\|\sum_{k=1}^{B} \omega_k^\Gamma \widehat{\mu}_k^\Gamma - \mu_1^\Gamma\Big\|_{\mathcal{H}_\Gamma}^2\Big] + C\sqrt{\frac{\log B}{D}}. \tag{56}$$

We can then apply Theorem 4.6 in the random Fourier features RKHS $\mathcal{H}_\Gamma$, so conditionally to $\Gamma$. For any subset $V \subset \llbracket B \rrbracket$:

$$\mathbb{E}_{\mathcal{D}}\Big[\Big\|\sum_{k=1}^{B} \widehat{\omega}_k^\Gamma \widehat{\mu}_k^\Gamma - \mu_1^\Gamma\Big\|_{\mathcal{H}}^2 \Big| \Gamma\Big] \le \Big[\Delta_{\Gamma,V}^2 + \frac{\mathrm{Tr}\,\Sigma_1^\Gamma + 2\Delta_{\Gamma,V}}{n_V}\Big] + \frac{Cu_0}{\sqrt{n_1}} \max\Big(\sqrt{\frac{|V|-1}{n_V}}, \frac{1}{\sqrt{n_1}}\Big) \max\Big(\frac{\mathrm{Tr}\,\Sigma_1^\Gamma}{\sqrt{d_1^{e,\Gamma}}}, \frac{u_0}{\sqrt{n_1}}\Big), \tag{57}$$

where $\Sigma_1^\Gamma$ is the covariance operator of the random Fourier features $\phi_\Gamma(Z)$ for $Z \sim \mathbb{P}_1$, $d_1^{e,\Gamma}$ is its effective dimension and $\Delta_{\Gamma,V} = \max_{k \in V}\|\mu_{\mathbb{P}_k}^\Gamma - \mu_{\mathbb{P}_1}^\Gamma\|_{\mathcal{H}_\Gamma}$.

Using Lemma E.6, we have $\mathbb{E}_\Gamma\big[\mathrm{Tr}\,\Sigma_1^\Gamma\big] = \mathrm{Tr}\,\Sigma_1$, and, using Jensen's inequality and Lemma E.7:

$$\mathbb{E}_\Gamma\Big[\frac{\mathrm{Tr}\,\Sigma_1^\Gamma}{\sqrt{d_1^{e,\Gamma}}}\Big] = \mathbb{E}_\Gamma\Big[\sqrt{\mathrm{Tr}\,\Sigma_1^\Gamma \|\Sigma_1^\Gamma\|_{op}}\Big] \le \sqrt{\mathrm{Tr}\,\Sigma_1 \mathbb{E}_\Gamma\big[\|\Sigma_1^\Gamma\|_{op}\big]} \le \sqrt{\mathrm{Tr}\,\Sigma_1}\sqrt{\|\Sigma_1\|_{op} + \sqrt{\frac{\mathrm{Tr}\,\Sigma_1}{D}}}.$$

Using that $\sqrt{a^2 + b} \le a + b/(2a)$ for $a, b \ge 0$, we obtain that:

$$\mathbb{E}_\Gamma\Big[\frac{\mathrm{Tr}\,\Sigma_1^\Gamma}{\sqrt{d_1^{e,\Gamma}}}\Big] \le \sqrt{\mathrm{Tr}\,\Sigma_1}\Big(\|\Sigma_1\|_{op}^{1/2} + C\sqrt{\frac{d_1^e}{D}}\Big) \le \frac{\mathrm{Tr}\,\Sigma_1}{\sqrt{d_1^e}} + C\sqrt{\frac{d_1^e}{D}}, \tag{58}$$

using that $\mathrm{Tr}\,\Sigma_1$ is upper bounded by 1 since the kernel is upper bounded by 1.

It remains to control $\Delta_{\Gamma,V}$. Using Lemma E.1 combined with an union bound over the $B$ agents, with probability at least $1 - e^{-u}$, for $u \ge 0$:

$$\Delta_{\Gamma,V} \le \max_{k \in V_\Delta}\|\mu_{\mathbb{P}_k} - \mu_{\mathbb{P}_1}\|_{\mathcal{H}} + C\sqrt{\frac{u \log B}{D}} = \Delta_V + C\sqrt{\frac{u \log B}{D}}, \tag{59}$$

where $\Delta_V = \sup_{k \in V} \mathrm{MMD}(\mathbb{P}_1, \mathbb{P}_k)$. It follows that

$$\mathbb{E}_\Gamma\Big[\Delta_{\Gamma,V}^2 + \frac{\mathrm{Tr}\,\Sigma_1^\Gamma + 2\Delta^\Gamma}{n_V}\Big] \le \Delta_V^2 + \frac{\mathrm{Tr}\,\Sigma_1 + 2\Delta_V}{n_V} + C\sqrt{\frac{\log B}{D}}.$$

using that $\Delta_V \le 1$. Combining the two upper bounds leads to the result. $\qquad\square$

