# OpenReview forum: "Adaptive Personalized Federated Learning via Multi-task Averaging of Kernel Mean Embeddings"
_ICML.cc/2026/Conference — ICML 2026 regular_

### Official Review · Reviewer_sCaz · 2026-03-09

**Soundness:** 2
**Presentation:** 2
**Significance:** 2
**Originality:** 2
**Overall Recommendation:** 3
**Confidence:** 4

**Summary:**

This paper proposes a novel personalized federated learning approach designed to address data heterogeneity, enabling multiple data agents to collaborate without sharing raw data. The method achieves this by adaptively learning cooperation weights between agents without relying on prior heterogeneity assumptions. The core innovation lies in combining Kernel Mean Embedding (KME) with multi-task learning to optimize the weighted empirical risk of agents, while quantifying the statistical gain of collaboration through Maximum Mean Discrepancy (MMD). To reduce communication costs, the paper also introduces an approximation method based on random Fourier features. Experimental results demonstrate that this approach effectively adapts to various types of data heterogeneity and outperforms existing methods across multiple tasks.

**Compliance With Llm Reviewing Policy:**

Affirmed.

**Key Questions For Authors:**

1.	This paper employs Kernel Mean Embedding (KME) and MMD-based metrics in its algorithms, which may pose challenges when selecting kernels. Improper kernel selection can lead to performance degradation. Consequently, the method's dependency on kernels constitutes one of its limitations, particularly across different application domains where selecting suitable kernels may require extensive experimental tuning.
2.	Although the paper provides experimental validation using synthetic data and the FEMNIST dataset, these datasets may not encompass all potential real-world application scenarios. In particular, the algorithm's performance may differ when applied to highly heterogeneous real-world data, necessitating further empirical research with broader application contexts.
3.	Although Q-aggregation methods have been proposed to optimize collaboration weights, the complexity of such optimization problems may limit their application in large-scale federated learning. Optimizing weights on massive datasets could become a bottleneck.

**Limitations:**

Yes

**Strengths And Weaknesses:**

1.	Unlike other methods, the algorithm presented in this paper does not rely on prior assumptions about agent heterogeneity and can adaptively learn cooperation weights from the data. This represents a core innovation of this paper, differing significantly from most existing personalized federated learning approaches, which typically require some form of structural assumption to guide the learning process.
2.	This paper proposes combining KME with mean estimation in multi-task learning to address the challenge of collaboration among heterogeneous agents, while also quantifying the statistical gain from such collaboration.

---

> ### Author Rebuttal · Authors · 2026-03-31
>
> We thank the reviewer for their feedback and answer to their questions below.
>
>
> **This paper employs Kernel Mean Embedding (KME) and MMD-based metrics in its algorithms, which may pose challenges when selecting kernels. Improper kernel selection can lead to performance degradation. Consequently, the method's dependency on kernels constitutes one of its limitations, particularly across different application domains where selecting suitable kernels may require extensive experimental tuning.**
>
> We agree that the choice of kernel is an important practical consideration. This limitation is inherent to kernel methods, to which KME belongs, and our approach is naturally sensitive to it.
>
> However, unlike classical kernel methods, we only assume that the loss function belongs to the RKHS, rather than requiring that $Y\simeq h(X)$ for some function $h$ in the RKHS. This is significantly less restrictive and reduces the need for extensive kernel tuning, as the kernel is used to capture similarities between distributions rather than to model the relationship between features and outputs. In practice, we recommend using universal kernels, such as the Gaussian kernel employed in our experiments, since such a kernel fully characterize distributions.
>
>
> **Although the paper provides experimental validation using synthetic data and the FEMNIST dataset, these datasets may not encompass all potential real-world application scenarios. In particular, the algorithm's performance may differ when applied to highly heterogeneous real-world data, necessitating further empirical research with broader application contexts.**
>
> While we agree with the reviewer that more experiments could be interesting, we can never "encompass all potential real-world application scenarios". We believe our current experiments cover a meaningful range of settings, including different types of distribution shifts (covariate shift and concept shift) and a standard personalized federated learning benchmark (FEMNIST). Moreover, we emphasize that the main contributions of our work are methodological and theoretical, supported by strong guarantees, which are effectively illustrated through these experiments.
>
> **Although Q-aggregation methods have been proposed to optimize collaboration weights, the complexity of such optimization problems may limit their application in large-scale federated learning. Optimizing weights on massive datasets could become a bottleneck.**
>
> On the contrary, we view the simplicity of weight computation via Q-aggregation as a key strength of our approach. In fact, the local optimization of the weights reduces to minimizing a quadratic form over the $B$-dimensional simplex (see Appendix A), which can be efficiently solved using standard first-order optimization methods. This optimization is typically much less complex than training the model parameter, especially when the number of agents $B$ is smaller that the model dimension, and it does not require communication during weight optimization.
>
> One might think that a potential bottleneck is the exchange of KMEs (vectors of size $D$) before weight optimization. However, $D$ is a free parameter, and as long as it is smaller than the model dimension, communicating KMEs remain less costly than exchanging model parameters, preserving the effiency of the overall method.
>
> We will clarify this in the revised version.

---

> > ### Author Rebuttal · Reviewer_sCaz · 2026-04-03
> >
> > Thanks for the response. I think that the authors have successfully addressed my concerns. I decided to increase the score to 4.

---

> > > ### Author Response · Authors · 2026-04-07
> > >
> > > We thank the reviewer for their feedback and for indicating an increase in their score. We kindly ask that this update be submitted before the deadline so it can be considered in the final decision. Thank you again for your time and effort in reviewing our paper.

---

### Official Review · Reviewer_bfXu · 2026-03-09

**Soundness:** 3
**Presentation:** 4
**Significance:** 4
**Originality:** 4
**Overall Recommendation:** 5
**Confidence:** 4

**Summary:**

The paper tackles the challenge of Personalized Federated Learning (PFL) by moving away from common structural assumptions, such as enforcing all local models to be close to a global model or assuming clients form a fixed number of clusters. Instead, the authors formulate the PFL objective—specifically, finding the optimal collaboration weights for a target client—as a high-dimensional mean estimation problem. By assuming the loss function resides within a Reproducing Kernel Hilbert Space (RKHS), they mathematically link the generalization error to the Maximum Mean Discrepancy (MMD) between a target agent's distribution and the estimated mixture of all agents' distributions.To solve this, they apply the Q-aggregation method to estimate the mixture weights by aggregating the Kernel Mean Embeddings (KMEs) of the agents' empirical distributions. To overcome the privacy and communication constraints inherent to Federated Learning (since sharing exact KMEs would theoretically require sharing raw data), the authors propose a practical approximation using Random Fourier Features (RFF). The paper provides finite-sample generalization guarantees on local excess risks and validates the method's adaptivity to data heterogeneity through experiments on synthetic (concept and covariate shift) and real-world (FEMNIST) datasets.
The originality of this work is a standout feature. Bridging high-dimensional multiple mean estimation—specifically utilizing the recent Q-aggregation method—with PFL via KMEs is a highly novel and creative connection. To the best of the authors' (and reviewer's) knowledge, establishing this formal connection is a first.

**Compliance With Llm Reviewing Policy:**

Affirmed.

**Key Questions For Authors:**

1.Optimization Error: Remark 2.1 explicitly ignores optimization error. How does imperfect optimization of the weighted empirical risk during the local model training phase impact your final excess risk bounds in practical, non-convex scenarios?
2.Communication Cost overhead: While sharing KMEs via RFF is stated to be communication-efficient relative to raw data , how does the upfront communication overhead of sharing $D$-dimensional vectors (where $D=500$ or $1000$ in your experiments) scale with $B$ clients, compared to standard personalized baselines that do not require this weight-learning preamble?
3.Assumption 4.1 limits: The assumption that the loss function lies in the RKHS is mathematically necessary for the proofs. How drastically does the method degrade empirically if this assumption is heavily violated (e.g., in deeper, more complex architectures than the evaluated 3-layer ReLU networks)?

**Limitations:**

No. While the authors briefly mention open questions in the Conclusion (such as quantifying privacy loss and approximating general loss functions), they do not have a dedicated Limitations section. Their "Impact Statement" is generic and dismissive ("none of which we feel must be specifically highlighted here").
Constructive Suggestion: The authors should be encouraged to add a dedicated "Limitations and Societal Impact" section. This should explicitly discuss the computational overhead of computing RFFs for clients with limited edge-device compute, as well as the precise privacy implications (e.g., lack of formal Differential Privacy guarantees) when sharing KMEs.

**Strengths And Weaknesses:**

Strengths
1.The submission is technically rigorous. Translating the PFL weighting problem into a KME aggregation problem is mathematically sound and cleverly executed. A major strength is the derivation of finite-sample generalization guarantees that explicitly quantify the statistical advantage of collaboration. This is a significant step up from many existing PFL methods that merely provide convergence guarantees on training data.
2.The paper is clearly written and well-structured. The narrative arc—moving from the idealized exact KME framework (Algorithm 1) to the practical, privacy-preserving RFF-based implementation (Algorithm 2)—provides a highly logical flow for the reader. The related work section is excellent, properly positioning the paper against existing meta-learning, cluster-based, and heuristic PFL approaches.
3.The paper addresses a highly relevant problem: handling non-IID data in FL without enforcing rigid, artificial structures on client relationships. The approach of sharing $D$-dimensional RFF approximations of KMEs instead of iteratively sharing model parameters during the collaboration weight-learning phase is a refreshing paradigm. This could influence future research into decoupling collaboration-graph learning from local model optimization.
4.The originality of this work is a standout feature. Bridging high-dimensional multiple mean estimation—specifically utilizing the recent Q-aggregation method—with PFL via KMEs is a highly novel and creative connection. To the best of the authors' (and reviewer's) knowledge, establishing this formal connection is a first.
Weaknesses
1.The theoretical analysis explicitly ignores the optimization error, focusing purely on statistical error (as noted in Remark 2.1). In practice, imperfect optimization of the weighted empirical risk could loosen the bounds. Additionally, Assumption 4.1 (that the loss belongs to an RKHS) is a relatively strong assumption. While the authors discuss universal kernels in Appendix D, the practical implications for highly complex non-convex deep learning losses remain a slight gap.
2. The notation becomes slightly heavy in the theoretical sections (e.g., Theorem 4.4 and Theorem 5.2). Furthermore, the transition between using kernels for concept shift (weighting the $Y$ component) versus covariate shift (kernels on $X$ alone) could be expanded slightly in the main text to ensure maximum clarity for general practitioners.

---

> ### Author Rebuttal · Authors · 2026-03-31
>
> We thank the reviewer for the positive feedback and detailed review. Below, we provide clarifications on the points that remain unclear and will update the paper to incorporate the suggested improvements.
>
> **Remark 2.1 explicitly ignores optimization error. How does imperfect optimization of the weighted empirical risk during the local model training phase impact your final excess risk bounds in practical, non-convex scenarios?**
>
> In general, the expected excess risk can be decomposed as the sum of two terms: the statistical error (right-hand side of Eq. 3), and the optimization error, which measures how well the weighted empirical risk $\hat{R}_\omega(\theta)$ is minimized. In convex settings, using FedAvg with stochastic gradients typically leads to an optimization error that decreases at a rate of $\mathcal{O}(\frac{1}{\sqrt{T}})$, which can be directly plugged as an additional error in Eq. (3).
>
> For non-convex functions, however, most gradient-based methods control optimization error in terms of convergence to stationary points (points with near-zero gradient), which does not directly bound the excess risk in Eq. (3). This is a general challenge in non-convex learning. Recent work has begun investigating excess risk control in terms of gradient norms [1]. While exploring this decomposition could be an interesting direction, we consider it beyond the scope of our current paper.
>
> [1] Yunwei Lei. Stability and Generalization of Stochastic Optimization with Nonconvex and Nonsmooth Problems. In Conference on Learning Theory (COLT), 2023
>
> **While sharing KMEs via RFF is stated to be communication-efficient relative to raw data, how does the upfront communication overhead of sharing $D$-dimensional vectors (where $D=500$ or $D=1000$ in your experiments) scale with $B$ clients, compared to standard personalized baselines that do not require this weight-learning preamble?**
>
> The communication of KMEs via RFFs has the advantage of occurring only once. If the dimension $D$ is on the order of the model size, this communication is comparable to sharing gradients or model updates, and is therefore negligible relative to training the full model.
>
> To further limit communication costs, weight learning can be performed via a central server, to which each agent transmits its KME (a vector of size $D$) and its covariance matrix ($D \times D$). The preferred strategy can be chosen depending on the number of agents $B$ and the feature dimension $D$, resulting in a total  communication cost of $B D\min( B,D)$.
>
> **The assumption that the loss function lies in the RKHS is mathematically necessary for the proofs. How drastically does the method degrade empirically if this assumption is heavily violated (e.g., in deeper, more complex architectures than the evaluated 3-layer ReLU networks)?**
>
> In response to your question, we ran the experiments from Section 6.2 using larger architectures (with 10 and 20 hidden layers). The results are consistent with those obtained with the 3-layer model: in the covariate shift setting, a sufficiently large model is able to fully learn the target function, and the GrandMean, Oracle, and Q-aggregation methods achieve comparable performance.
>
> In fact, for a universal kernel, increasing model complexity *does not* heavily violate Assumption 4.1. Indeed, even for complex models, the associated loss function remains continuous (and bounded for bounded data), and can therefore be approximated by functions in the RKHS. As noted by the reviewer, these properties are discussed in Section 5.2 and Appendix D. The effect of increasing the model's complexity appears in our theoretical bounds through the terms $R_\Theta$ and $R_{\Theta,\varepsilon}$.
>
> Lasty, we would like to emphasize that weight learning with Q-aggregation is completely independent of the model architecture. Hence, increasing the model complexity will not affect the weight learning, which will essentially make sure that the distibution (the KME) of the target agent is well estimated for the chosen MMD. Using universal kernels, we expect this MMD distance to be relevant for any architectures.
>
> We will incorporate these new experiments and a dedicated discussion of these considerations in the revised version, as suggested by the reviewer.

---

### Official Review · Reviewer_MLiD · 2026-03-13

**Soundness:** 3
**Presentation:** 4
**Significance:** 3
**Originality:** 3
**Overall Recommendation:** 5
**Confidence:** 3

**Summary:**

The authors reduce the "all-for-one" personalized federated learning problem to a high-dimensional mean estimation problem, and propose a solid approach for leveraging other datasets while only communicating an *average* of the kernel-mean embeddings. To get around the problem of computing all pairwise embeddings of datasets, they use RFFs.

**Compliance With Llm Reviewing Policy:**

Affirmed.

**Final Justification:**

Reviewer addressed my concerns, which were minor, so the score remains the same.

**Key Questions For Authors:**

How do you plan to assess the quality of the loss function approximation when you diverge from the "universal" case with RFFs?

How will this work in practice with highly heterogeneous (in type not just in data populations) medical data?

I found the experiment in Section 6.2 really confusing. Can you explain this experiment in much more detail?

**Limitations:**

Yes

**Strengths And Weaknesses:**

The approach is an original reformulation of the problem and the theoretical justification is solid. The submission is in great shape, with only a few scattered minor usage issues. The work left out (choosing the number of RFFs for, on the low end, low bandwidth needs and stronger privacy guarantees, and on the high end, better approximation of the loss function; choosing appropriate kernels; experimental validation in settings most ) is, in my view, appropriate to leave for future work. The main drawback is the lack of experiments on datasets more relevant for the federated learning setting. The significance may be ultimately limited in practice but this will be more difficult to establish.

---

> ### Author Rebuttal · Authors · 2026-03-31
>
> We thank the reviewer for the positive feedback and for raising several interesting questions.
>
> **How do you plan to assess the quality of the loss function approximation when you diverge from the "universal" case with RFFs?**
>
> We are not entirely sure we fully understand the reviewer's question or what is meant by “diverge.” In particular, two types of approximations appear in our work: (1) the kernel approximation via RFFs, used to obtain a practical federated algorithm (Section 5.1), and (2) the approximation of the loss by an universal kernel, used to relaxate Assumption 4.1 and discussed in Section 5.2 and Appendix D.
>
> The RFF approximation is explicitly quantified in Theorem 5.2. The approximation due to the choice of kernel is controlled via Assumption 4.1, which requires the loss function to lie in the RKHS. To satisfy this assumption, we use universal kernels, such as the Gaussian kernel, whose RKHS has strong approximation properties. The theoretical cost of this approximation is detailed in Appendix D: Lemma D.1 provides a framework for evaluating how well the RKHS can approximate the loss function, using the complexity measure $R_{\Theta,\varepsilon}$ (Eq. 29). One potential direction for more explicit results is to quantify the dependence of $R_{\Theta,\varepsilon}$ on $\varepsilon$ and optimize bound (28).
>
> We hope this clarifies the different approximations made in our paper and their theoretical implications. We are, of course, happy to provide further details if needed.
>
> **How will this work in practice with highly heterogeneous (in type not just in data populations) medical data?**
>
> Thank you for this very interesting question. Our framework currently applies only when the data are of the same type across all agents, which is a standard assumption in federated learning. Handling data of different types introduces additional challenges that are largely orthogonal to the focus of our work.
>
> That said, if a common embedding space can be constructed, our method could be applied directly by comparing the resulting representations. Alternatively, one could rely on features shared across agents to compute the weights. A careful treatlent of such scenarios, however, is beyond the scope of this paper.
>
> **I found the experiment in Section 6.2 really confusing. Can you explain this experiment in much more detail?**
>
> We apologize for the confusion and clarify the experimental setting here.
>
> In this section, we evaluate the performance of our method when under heterogeneity arising solely from a covariate shift. Specifically, the target is $Y = f(X)$, where the function $f$ is the same for all agents, but the feature distributions $X$ vary across agents. At first glance, personalization might appear unnecessary under covariate shift, since a model perfectly learning $f$ would generalize to all agents. However, when the model is not sufficiently complex, it cannot fully capture $f$. In this case, personalization helps by selecting relevant collaborators with similar distributions.
>
> Our experiment is designed to illustrate this effect. We define three types of agents, as detailed in Equation (26): agents of the same type have similar feature distributions, while distributions differ across types. We focus on learning a model for an agent of type 1 and test several architectures. Figure 2 illustrates the above phenomenon. For small models, training on all agents (GrandMean) performs worse than local training, because using only local data allows the model to focus on the portion of the feature space actually observed by the agent. For larger models, which can learn $f$ over the entire space, GrandMean performs better. Our personalization method consistently improves learning by selecting only the relevant agents, benefiting both small and large models, and performs nearly as well as an oracle trained only on type 1 agents.

---

> > ### Author Rebuttal · Reviewer_MLiD · 2026-04-06
> >
> > Thanks for the clarifications. I will keep the score at 5 (accept).

---

### Official Review · Reviewer_7BhM · 2026-03-13

**Soundness:** 3
**Presentation:** 3
**Significance:** 3
**Originality:** 3
**Overall Recommendation:** 4
**Confidence:** 4

**Summary:**

The paper proposed a new method for personalized federated learning where each agent optimizes a weighted combination of all agents' local loss functions. The authors proposed a kernel-based method to learn the weights from data by formulating the estimation of these
weights as a kernel mean embedding problem.  The paper derives finite-sample guarantees on local excess risks under certain assumptions.

**Compliance With Llm Reviewing Policy:**

Affirmed.

**Key Questions For Authors:**

Please see weaknesses above.

**Limitations:**

Yes.

**Strengths And Weaknesses:**

**Strengths.** The paper is well-written and easy to follow. The presented approach and results are novel and intuitive. The technical contribution of the paper is interesting and useful for FL researchers.

**Weaknesses.**

- In the abstract and the title, the authors mention that their algorithm is adaptive in the sense that, without the knowledge of data heterogeneity, the proposed algorithm can automatically transition between global and local learning regimes. I did not see any discussion of this fact in the main content of the paper. According to the discussion after Thm 4.4, it seems that to switch between local and global regimes, knowledge of $\Delta_V$ is required at the local node? Is this knowledge trivially available at local nodes? Please add a discussion in the paper, and also explain if this is feasible.

- Assumption 4.1 demands a specific structure from the loss function, which limits the applicability of the developed methods. Can the authors provide more practice examples where the proposed assumption on the loss function is satisfied?

- In Algorithm 2, each agent exchanges a high-dimensional computed mean embedding based on its local data. Can this lead to any privacy issues?

- Since each agent computes a different set of weights, will this add another level of difficulty in implementing the FL algorithm compared to the classical setting, where each agent solves the FL problem with the same set of weights? Please discuss the differences.

- The experiments are conducted on relatively small scale prolems; it may be interesting to see the applicability of the proposed approach on large-scale FL problems.

---

> ### Author Rebuttal · Authors · 2026-03-31
>
> We thank the reviewer for the positive feedback and provide answer to their questions below.
>
> **Automatic transition between global and local learning regimes and knowledge of $\Delta_V$**
>
> Thank you for raising this point. Adaptivity is a key feature of our approach, and we want our contribution to be clearly understood in this regard. To clarify, the bound on the MMD distance in Eq. 14 of Theorem 4.4 is an oracle bound: the inequality holds for any choice of set $V$, or equivalently, the MMD distance is bounded by the minimum over all possible subsets $V$ of agents. In other words, **the resulting error is no larger than the error obtained from the best aggregation over any subset $V$ of agents**. In particular, the resulting error is smaller than that of local training ($V=\{1\}$, see Example 4.5) or global training ($V=\{1,\dots,B\}$). Crucially, this is achieved **without any prior knowledge on the agent's distributions**, which is why we describe the method as adaptive. More specifically, **neither $\Delta_V$ nor any pairwise distance between agents needs to be known**.
>
> To clarify this important result, we will add the following sentences to the discussion under Theorem 4.4:
>
> "Note that while our method automatically optimizes the upper-bound in (14), it does not require knowledge of the theoretical quantities involved, such as $\Delta_V$ or $\Sigma_1$, highlighting its adaptivity. Overall, our method fully adapts to varying degrees of heterogenity, seamlessly transitioning from the homogeneous regime (where it recovers the rate of a single global training) to the highly heterogenous regime (where it recovers the rate of purely local training)."
>
> **Applicability of Assumption 4.1**
>
> In practice, we recommend using universal kernels, such as the Gaussian one used in our experiments. These kernels have the property that any bounded continuous function can be approximated to arbitrary precision by a function in their RKHS (see Section 5.2 and Appendix D for details).
>
> The use of complex models (and, consequently, complex loss functions) affects the theoretical bounds via the quantity $R_\Theta$ or $R_{\Theta,\varepsilon}$), but it does not hinder practical application, since the learning of the weights is decoupled from model training. For example, in Section 6.2, we trained neural networks of varying complexity using weights learned with a Gaussian kernel and consistently observed significant improvements over both local and centralized training (see Figure 2).
>
> **Privacy considerations of sharing kernel mean embeddings**
>
> While sharing high-dimensional mean embeddings could potentially raise privacy concerns, we have not quantified this risk, as our focus is on the personalization process and its practical implementation using random Fourier features. A formal evaluation of privacy loss is left for future work, as noted in the conclusion.
>
> We can, however, offer some general observations. Intuitively, the dimension $D$ of the random Fourier features affects the potential privacy loss: if $D$ is comparable to the model size, the risk is similar to that of sharing gradients or model updates in standard federated training. Moreover, since Kernel Mean Embeddings are averages of bounded functions over the dataset, they could likely be effectively privatized by adding small amounts of noise.
>
> **Implications of per-agent weights on the implementation of personalized FL**
>
> We thank the reviewer for this question. In the classical FL setting, a single parameter vector $\theta$ is learned for all agents. In contrast, in personalized FL, each agent has its own parameter vector.
>
> The simplest (though naive) approach is to run the optimization algoritm $B$ times (once per agent). This is what we used in the experiments, resulting in a computational cost roughly $B$ times higher than in classical FL. However, the $B$ runs can be executed in parallel, and at each communication round, the messages from all runs can be batched into a single transmission, reducing communication overhead.
>
> Looking ahead, we plan to explore simultaneous learning of all agents' parameter vectors, where exchanged gradients are computed at different parameter vectors. In theory, this would only impact the optimization error, requiring new convergence analysis, but not our generalization error analysis.
>
> **Scale of experiments**
>
> We agree that larger-scale FL experiments would be interesting. Nevertheless, our experimental setups are non-trivial: all experiments involve roughly one hundred agents, with data dimensionality up to 782 (as in FEMNIST), and each agent has only a limited number of samples (on the order of tens), corresponding precisely to the regime where personalized FL is most beneficial. Finally, we emphasize that the main contributions of our work are methodological and theoretical, supported by strong guarantees, which are effectively illustrated through these experiments.

---

> > ### Author Rebuttal · Reviewer_7BhM · 2026-04-04
> >
> > n/a

---

### Decision · Program_Chairs · 2026-04-30

**Decision:**

Accept (regular)

**Comment:**

This paper presents a new method for personalized federated learning via multi-task averaging of kernel mean embeddings. It also derives finite-sample guarantees on local excess risks under certain assumptions. Overall, this is a good paper with clear contribution which is appreciated by all reviewers. I concur with the reviewers and recommend acceptance.